# Mobilome-driven segregation of the resistome in biological wastewater treatment

**Laura de Nies[1], Susheel Bhanu Busi[1], Benoit Josef Kunath[1], Patrick May[2], Paul Wilmes[1,3]***

[1]Systems Ecology Group, Luxembourg Centre for Systems Biomedicine, University of Luxembourg, Esch-sur-Alzette, Luxembourg; [2]Bioinformatics Core, Luxembourg Centre for Systems Biomedicine, University of Luxembourg, Esch-sur-Alzette, Luxembourg; [3]Department of Life Sciences and Medicine, Faculty of Science, Technology and Medicine, University of Luxembourg, Esch-sur-Alzette, Luxembourg

***For correspondence:**
paul.wilmes@uni.lu

**Competing interest:** The authors declare that no competing interests exist.

**Abstract** Biological wastewater treatment plants (BWWTP) are considered to be hotspots for the evolution and subsequent spread of antimicrobial resistance (AMR). Mobile genetic elements (MGEs) promote the mobilization and dissemination of antimicrobial resistance genes (ARGs) and are thereby critical mediators of AMR within the BWWTP microbial community. At present, it is unclear whether specific AMR categories are differentially disseminated via bacteriophages (phages) or plasmids. To understand the segregation of AMR in relation to MGEs, we analyzed meta-omic (metagenomic, metatranscriptomic and metaproteomic) data systematically collected over 1.5 years from a BWWTP. Our results showed a core group of 15 AMR categories which were found across all timepoints. Some of these AMR categories were disseminated exclusively (bacitracin) or primarily (aminoglycoside, MLS and sulfonamide) via plasmids or phages (fosfomycin and peptide), whereas others were disseminated equally by both. Combined and timepoint-specific analyses of gene, transcript and protein abundances further demonstrated that aminoglycoside, bacitracin and sulfonamide resistance genes were expressed more by plasmids, in contrast to fosfomycin and peptide AMR expression by phages, thereby validating our genomic findings. In the analyzed communities, the dominant taxon *Candidatus* Microthrix parvicella was a major contributor to several AMR categories whereby its plasmids primarily mediated aminoglycoside resistance. Importantly, we also found AMR associated with ESKAPEE pathogens within the BWWTP, and here MGEs also contributed differentially to the dissemination of the corresponding ARGs. Collectively our findings pave the way toward understanding the segmentation of AMR within MGEs, thereby shedding new light on resistome populations and their mediators, essential elements that are of immediate relevance to human health.

## Editor's evaluation

This paper reports important results regarding the presence and potential dissemination of antibiotic resistance genes in wastewaters by convincingly combining analysis of gene abundance, expression, and association with mobile genetic elements and bacterial taxa. Via systematic evaluation and implementation of multiple tools, the authors provide a valuable approach for monitoring antibiotic resistance genes in the environment and assessing their dispersal and possible risks to human health.

## Introduction

Throughout human history, bacterial infections have been a major cause of both disease and mortality (*Bonilla and Muniz, 2009*). The discovery as well as the subsequent development and medical use of antibiotics have provided effective treatment options which limited the development and spread of bacterial pathogens. However, the use of antibiotics has exacerbated the emergence of antimicrobial resistance (AMR) in both commensal and pathogenic bacteria (*Wright, 2007*). As a result, AMR, as the 'silent pandemic', has become a prevalent threat to human health (*Brogan and Mossialos, 2016*; *Mahoney et al., 2021*; *O'Neill, 2014*).

From a public health perspective, biological wastewater treatment plants (BWWTPs) are considered hotspots of AMR due to the convergence of antibiotics with resistant, potentially pathogenic microorganisms originating from both the general population as well as agriculture, healthcare services and industry (*Alexander et al., 2020*; *Rodríguez-Molina et al., 2019*). Additionally, the mobilization of antimicrobial resistance genes (ARGs) through rampant horizontal gene transfer (HGT) promotes the dissemination of AMR within the BWWTP microbial community (*von Wintersdorff et al., 2016*). Therefore, BWWTPs represent an environment exceptionally suited for the evolution and subsequent spread of AMR (*Calero-Cáceres et al., 2014*; *Chen et al., 2013*). To date, more than 32 studies have documented the role of BWWTPs as key reservoirs of AMR (*Fouz et al., 2020*). Furthermore, BWWTPs generally do not contain the necessary infrastructure to remove either ARGs or resistant bacteria, which are released into the receiving water via the effluent, promoting its spread in the environment at large (*Alexander et al., 2020*). Most often these are surface water bodies such as rivers, which contribute to the further dissemination of AMR and resistant bacteria among environmental microorganisms (*Singer et al., 2016*). Acquired resistance may in turn be carried over to humans and animals using these water resources. In fact, there is strong evidence suggesting that ARGs from environmental bacteria can be taken up by human-associated and pathogenic bacteria (*Nadeem et al., 2020*; *Trinh et al., 2018*). From an epidemiological and surveillance perspective, BWWTPs also provide samples representative of entire populations (*Hendriksen et al., 2019*). As such, BWWTPs have recently been crucial for the monitoring of SARS-CoV-2 within the human population (*Herold et al., 2021*). Overall, to increase our understanding of the dissemination of AMR and the underlying mechanisms as well as its general prevalence, it is necessary to map the resistome of various environments starting with biological BWWTPs because it is critical to unravel the extent to which they act as reservoirs for the dissemination of antimicrobial resistance genes (ARGs) to bacterial pathogens. Moreover, understanding the community-level overviews of the ARG potential and its expression, coupled with population-level linking, including to pathogens, may allow for efficient monitoring of pathogenic and AMR potential with broad impacts on human health.

The presence of resistance genes and mobile genetic elements (MGEs) along with sub-inhibitory antibiotic selection pressures may facilitate HGT of ARGs into new hosts through the mobilome (*von Wintersdorff et al., 2016*). Previous work has in particular shown that antibiotic selection pressures may alter HGT processes, thereby increasing the number of resistance elements which reside on mobile DNA (*Datta and Hughes, 1983*). Acquisition of ARGs via MGEs primarily occurs through two mechanisms: conjugation or transduction (*MacLean and San Millan, 2019*). In conjugation, plasmids carrying one or more resistance genes are transferred between microorganisms (*Carattoli, 2013*), while in transduction bacteriophages carrying ARGs infect bacteria and integrate their genome into those of the host thereby conferring resistance (*Chiang et al., 2019*). Of these mechanisms, conjugation is often thought to have the greatest influence on the dissemination of ARGs, while transduction is deemed less important (*von Wintersdorff et al., 2016*). In general terms, studies concerning AMR and its dissemination focus either on phage (*Lood et al., 2017*; *Strange et al., 2021*) or plasmids solely (*Li et al., 2019*). Alternatively, the two are treated collectively (*Alexander et al., 2020*; *Che et al., 2019*) without a comprehensive comparative analysis. This circumstance has created a knowledge gap whereby the contributions of plasmids and phages as independent entities to AMR transmission within complex communities, such as those found in biological BWWTPs, is largely unknown.

To shed light on the dissemination and potential segregation of AMR within MGEs in a WTTP microbial community, we leveraged longitudinal meta-omics data (metagenomics, metatranscriptomics, and metaproteomics). Samples collected for 51 consecutive weeks over a period of 1.5 years were used to characterize the resistome. We found that several bacterial orders such as Acidimicrobiales, Burkholderiales, and Pseudomonadales were associated with 29 AMR categories across all timepoints. Our

longitudinal analysis suggests that MGEs are important drivers of AMR dissemination within BWWTPs. More importantly, we reveal that MGEs, that is plasmidomes and phageomes, contribute differentially to AMR dissemination. Furthermore, we observed this phenomenon in clinically-relevant taxa such as the ESKAPEE pathogens (*Reza et al., 2019*), for which plasmids and phages were exclusively associated with specific ARGs. Collectively, our data suggest that BWWTPs are critical reservoirs of AMR which show clear evidence for the segregation of distinct AMR genes within MGEs especially in complex microbial communities. In general, we believe that these findings may provide crucial insights into the segregation of the resistome via the mobilome in any and all reservoirs of AMR, including but not limited to animals, humans, and other environmental systems.

## Results

### Longitudinal assessment of the resistome within a BWWTP

To characterize the BWWTP resistome, we sampled a municipal BWWTP on a weekly basis over a 1.5-year period (ranging from 21-03-2011 to 03-05-2012) (*Herold et al., 2020*; *Martínez Arbas et al., 2021*). Metagenomic and metatranscriptomic reads were preprocessed, and both sets of reads co-assembled using the Integrated Meta-omic Pipeline as described previously (*Narayanasamy et al., 2016*). Subsequently, utilizing the PathoFact pipeline (*de Nies et al., 2021*) on the assembled contigs (*Methods*), we resolved the BWWTP resistome. This analysis revealed the presence of 29 different categories of AMR within the BWWTP. Subsequent longitudinal analyses highlighted enrichments in aminoglycoside, beta-lactam and multidrug resistance genes (*Figure 1a*). Concomitantly, we observed specific shifts in the AMR profiles over time. For example, a transient change at two timepoints (13-05-2011, 08-02-2012) highlighted a steep increase in resistance genes corresponding to glycopeptide resistance. Other AMR categories, such as diaminopyrimidine resistance, exhibited a less drastic but more fluid change in longitudinal abundance observable over multiple timepoints.

Additionally, AMR categories were found to persist over time within the BWWTP (*Figure 1b*). A core group of 15 AMR categories in total were identified and found to be present across the 1.5-year sampling period. These included aminoglycoside, beta-lactam, and multidrug resistance genes, which contributed the most to the pool of ARGs. A further six (aminocoumarin, aminoglycoside:aminocoumarin, elfamycin, nucleoside, triclosan, and unclassified) AMR categories were found to be prevalent (>75% of all timepoints), while another three AMR categories were moderately (50–75% of all timepoints) present over time (*Figure 1b*). Five other categories were rarely present within the BWWTP, with resistance corresponding to acridine dye only present at six of the timepoints. Altogether, this emphasized that the BWWTP resistome varies over time, substantiating the requirement for a longitudinal analysis to obtain an accurate overview of the community's overall resistome.

Although the data thus far provided a clear overview of the BWWTP from a metagenomic perspective, it did not provide any information regarding AMR expression. We therefore utilized the corresponding metatranscriptomic dataset to investigate the expression of identified ARGs and monitor their changes, within the BWWTP, over time. In contrast to the metagenomic data, we observed a difference in AMR expression levels for several categories. Aminoglycoside, beta-lactam, and multidrug resistance identified at high levels in metagenomic information were also highly expressed within the BWWTP (*Figure 1c*). However, peptide resistance demonstrated the highest expression levels of all the AMR categories. We further investigated which ARG subtypes contributed to the identified peptide resistance category and found that ~90% of the expressed peptide resistance was directly contributed by a single resistance gene, *YojI*, which was found to be widely distributed among the major taxa comprising the BWWTP community such as the Comamonadaceae (*Figure 1—figure supplement 1*). YojI is typically associated with resistance to microcins by reducing the intracellular concentration of the toxic antibiotic peptide (*Delgado et al., 2005*). The high incidence of this gene indicates a broad adaptive strategy amongst the microbial populations in the BWWTP against these specific stressors.

### Microbial community and co-occurrence patterns of AMR

Based on the previously identified microbial community (*Herold et al., 2020*), we hypothesized that the abundant and prevalent bacterial orders such as Acidimicrobiales were major contributors to the abundance in ARGs observable via metagenomics. To further investigate the contribution to AMR by

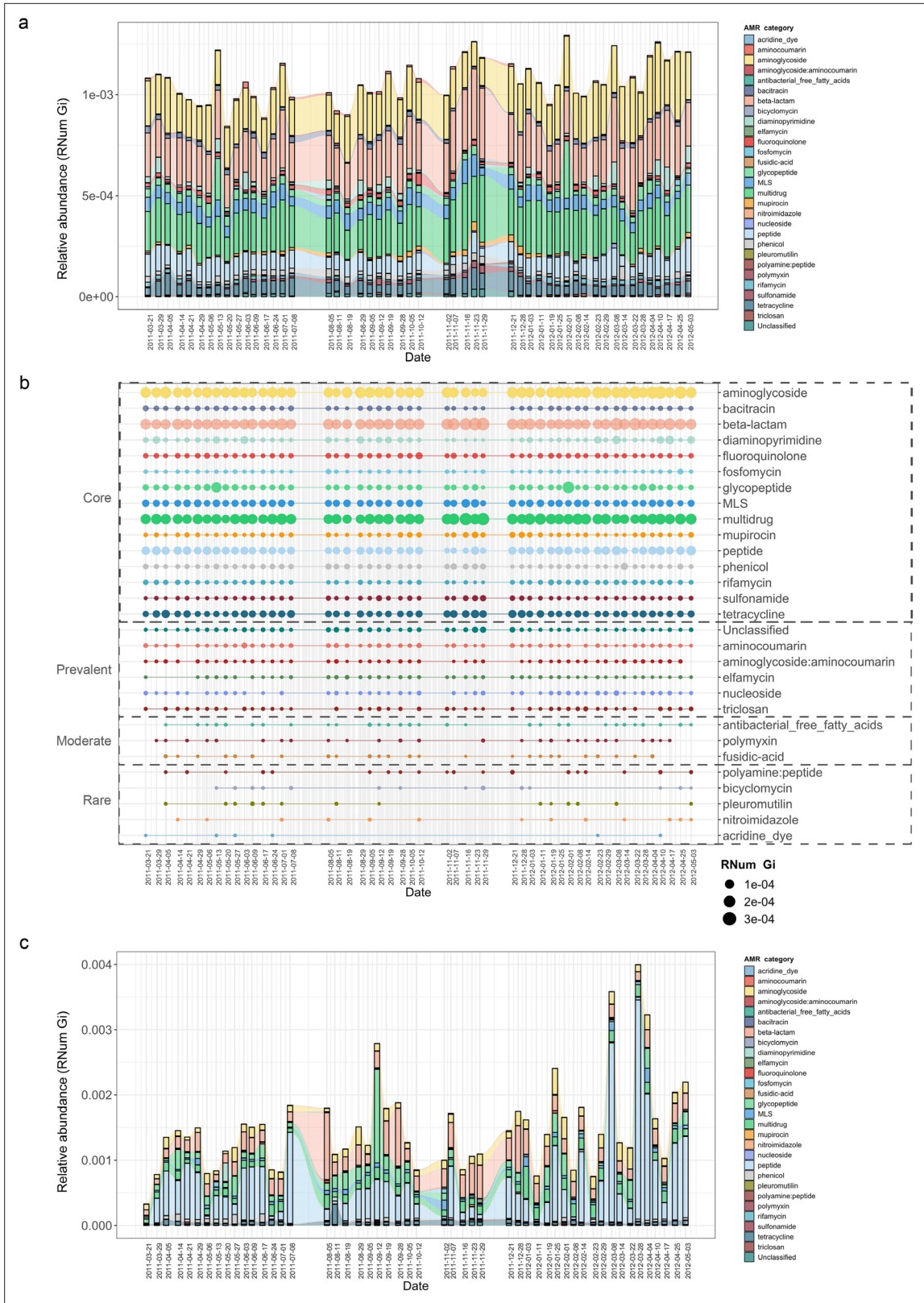

**Figure 1.** Longitudinal metagenomic and metatranscriptomic assessment of AMR. (**a**) ARG relative abundances over time within the BWWTP. (**b**) AMR categories at various timepoints categorized in four distinct groups based on presence/absence: Core (all timepoints), Prevalent (>75% of timepoints), Moderate (50–75% of timepoints), and Rare (<50% of all timepoints). (**c**) Relative abundance levels of expressed AMR categories over time within the BWWTP. Colors of all panels correspond to the AMR categories.

*Figure 1 continued on next page*

*Figure 1 continued*

The online version of this article includes the following figure supplement(s) for figure 1:

**Figure supplement 1.** Expression levels of individual ARGs.

the distinct microbial populations, we linked AMR genes to the contig-based taxonomic annotations of the assemblies (Methods). Herein, we identified a wide variety of taxonomic orders contributing to AMR, with multiple orders often contributing to the same resistance categories (*Figure 2—figure supplement 1*). Overall, taxa belonging to Acidimicrobiales, followed by Burkholderiales, were found to encode most of the ARGs (*Figure 2a*). Additionally, the abundance of ARGs linked to taxonomy varied over time. This was most noticeable during a five-week period (autumn: 02-11-2011 to 29-11-2011), where a decrease in abundance in ARGs linked to Acidimicrobiales and Bacteroidales was observed coinciding with an increase in ARG abundance in Pseudomonadales and Lactobacillales.

Since the order Acidimicrobiales was found to be linked to the highest abundance in ARGs, we further resolved the taxonomic affiliation and identified the species *Candidatus* Microthrix parvicella (hereafter known as *M. parvicella*) to be the main contributor to AMR. *M. parvicella* was previously found to dominate this microbial community (*Martínez Arbas et al., 2021*) and is a well-characterized bacterium commonly occurring in the BWWTP (*Calusinska et al., 2018*). Overall, aminoglycoside, beta-lactam, multidrug, and peptide resistance were found to be abundant in this species (*Figure 2b*), with aminoglycoside resistance demonstrating the highest expression levels as confirmed through metatranscriptomic analysis (*Figure 2—figure supplement 2*). Although it was not surprising to find a high abundance of ARGs linked to this species, the longitudinal variation in the abundances of these ARGs was nevertheless surprising (*Figure 2b*). Furthermore, coupled to a decrease in the abundance of *M. parvicella* itself (*Martínez Arbas et al., 2021*), we observed an almost complete decrease in ARGs at two timepoints (23-11-2011 and 29-11-2011). However, the *M. parvicella* population recovered to levels resembling the earlier timepoints in conjunction with the abundances in ARGs toward the end of the sampling period (*Figure 2a*, *Figure 2b*), underlining their overall contribution to AMR within this BWWTP. Alternatively, it is plausible that the dominance of *M. parvicella* is attributable to the encoded ARGs, which in turn, may confer a fitness advantage.

In order to determine whether the abundances in ARGs may be directly associated with the community composition over time, co-occurrence patterns between ARG subtypes and taxa (genus level) were explored using the metagenomic data. Association network analyses (*Figure 2c*) demonstrated that ARGs, within or across ARG types and microbial taxa, showed clear and distinct co-occurrence patterns within the BWWTP. These patterns indicated a strong segregation of distinct, taxa-specific ARG subtypes within the BWWTP community over time. One clear example was that of *M. parvicella* which encoded different aminoglycoside resistance genes (*Figure 2c*). Thus, the abundance of this bacterium along with the aminoglycoside ARGs were highly correlated.

## Monitoring pathogenic microorganisms within BWWTPs

In conjunction with the families observed within BWWTPs, we also found that certain ESKAPEE pathogens (*Reza et al., 2019*), such as *Klebsiella* spp. and *Pseudomonas* spp., demonstrated co-occurring patterns with ARGs (*Figure 2c*).

As previously mentioned, BWWTPs represent a collection of potentially pathogenic microorganisms originating from, among others, the human population. Moreover, evidence suggests that ARGs from environmental and commensal bacteria can spread to pathogenic bacteria through HGT (*MacLean and San Millan, 2019*). Therefore, we assessed the presence of AMR in the extended priority list of pathogens (*Table 1*), characterized as such by the WHO (*Tacconelli et al., 2018*), using both metagenomics and metatranscriptomics.

Of the identified pathogens (*Table 1*), we found that *Pseudomonas aeruginosa*, both encoded and expressed the highest abundance of ARGs, followed by *Acinetobacter baumannii*, over time within the BWWTP (*Figure 3*). Moreover, an increase in ARG abundance and expression was observed in *Pseudomonas aeruginosa* during the time period, during which the otherwise dominant *M. parvicella* demonstrated reduced abundance (*Figure 2b* and *Figure 3*).

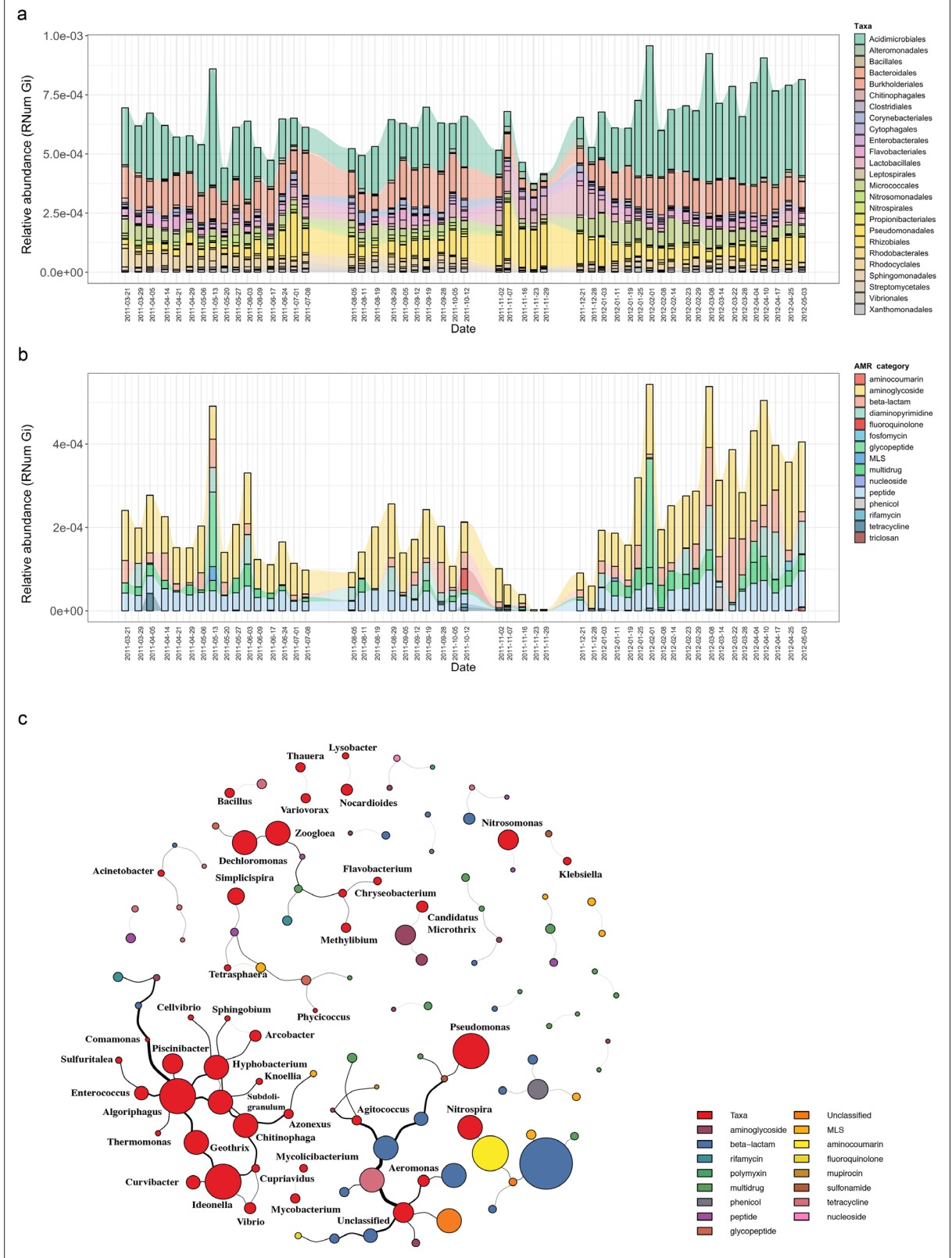

**Figure 2.** Microbial population-linked AMR. (**a**) Longitudinal ARG relative abundance levels linked to their corresponding microbial taxa (order level). Colors correspond to AMR categories. (**b**) Relative abundance of AMR categories linked to *Candidatus* Microthrix parvicella with colors corresponding to AMR categories. (**c**) Association network depicting co-occurrence patterns of individual antimicrobial resistance genes (ARGs) and microbial taxa

*Figure 2 continued on next page*

*Figure 2 continued*

on genus level. Nodes represent taxa or ARG with the node size representing the number of edges. The size of the edges represent the strength of interaction between the nodes.

The online version of this article includes the following figure supplement(s) for figure 2:

**Figure supplement 1.** Taxonomic diversity of AMR.

**Figure supplement 2.** Expressed AMR categories of *Candidatus* Microthrix parvicella.

## Differential transmission of antimicrobial resistance via mobile genetic elements

As previously described (*Beceiro et al., 2013*; *Wee et al., 2020*), the mobilome is a major contributor to the dissemination of AMR within a microbial community. Consequently, to understand (i) the role of MGE-mediated AMR transfer within the BWWTP, and (ii) to identify differential contribution of the mobilome to the dissemination of AMR, we identified both plasmids and phages within the metagenome and linked these to the respective ARGs. While, as expected, the majority of ARGs was found to be encoded on the bacterial chromosome (*Figure 4—figure supplement 1*), we also found that plasmids contributed to an average of 10.8% of all ARGs, while phage contributed to an average of 6.8% of all resistance genes, in agreement with the general hypothesis that conjugation has the greatest influence on the dissemination of ARGs (*adj.p*<0.05, One-way ANOVA) (*MacLean and San Millan, 2019*). This phenomenon, however, varied across time within the BWWTP (*Figure 4a*).

When investigating the dissemination of AMR via MGEs, most reports typically focus on either phages or plasmids individually, or both as collective contributors to transmission (*Slizovskiy et al., 2020*). To date and to our knowledge, the respective contributions of phages and plasmids to AMR transmission have not been subjected to a comprehensive comparative analysis. To facilitate a systematic, comparative view of MGE-mediated AMR, we assessed the segregation of MGEs with respect to AMR and found that phages and plasmids contributed differentially to AMR (*Figure 4—figure supplement 2*). Specifically, we tested 28 AMR categories with respect to their association with MGEs and found a significant difference in six AMR categories when comparing ARGs encoded by phages and plasmids (*adj.p* <0.05, Two-way ANOVA)(*Figure 4b*). Aminoglycoside, bacitracin, MLS (i.e. macrolide, lincosamide, and streptogramin) and sulfonamide resistance were found to be primarily encoded by plasmids, whereas fosfomycin and peptide resistance were found to be associated with phages.

To further understand AMR in relation to the community dynamics, we investigated the abundance and segregation of the above-mentioned significant resistance categories at different timepoints within the BWWTP. We observed ARG abundances varied over time both in phages (*Figure 4c*) as

**Table 1.** WHO priority list for research and development of new antibiotics for antibiotics-resistant bacteria (*Tacconelli et al., 2018*).

| Bacteria | Priority | Organism detected | Resistance detected |
|---|---|---|---|
| *Acinetobacter baumannii* | Critical | + | + |
| *Pseudomonas aeruginosa* | Critical | + | + |
| *Enterobacteriaceae* | Critical | + | + |
| *Enterococcus faecium* | High | + | + |
| *Staphylococcus aureus* | High | + | + |
| *Helicobacter pylori* | High | + | + |
| *Campylobacter* spp | High | + | - |
| *Salmonella* spp | High | + | + |
| *Neisseria gonorrhoeae* | High | + | - |
| *Streptococcus pneumoniae* | Medium | + | + |
| *Haemophilus influenzae* | Medium | + | - |
| *Shigella* spp | Medium | + | + |

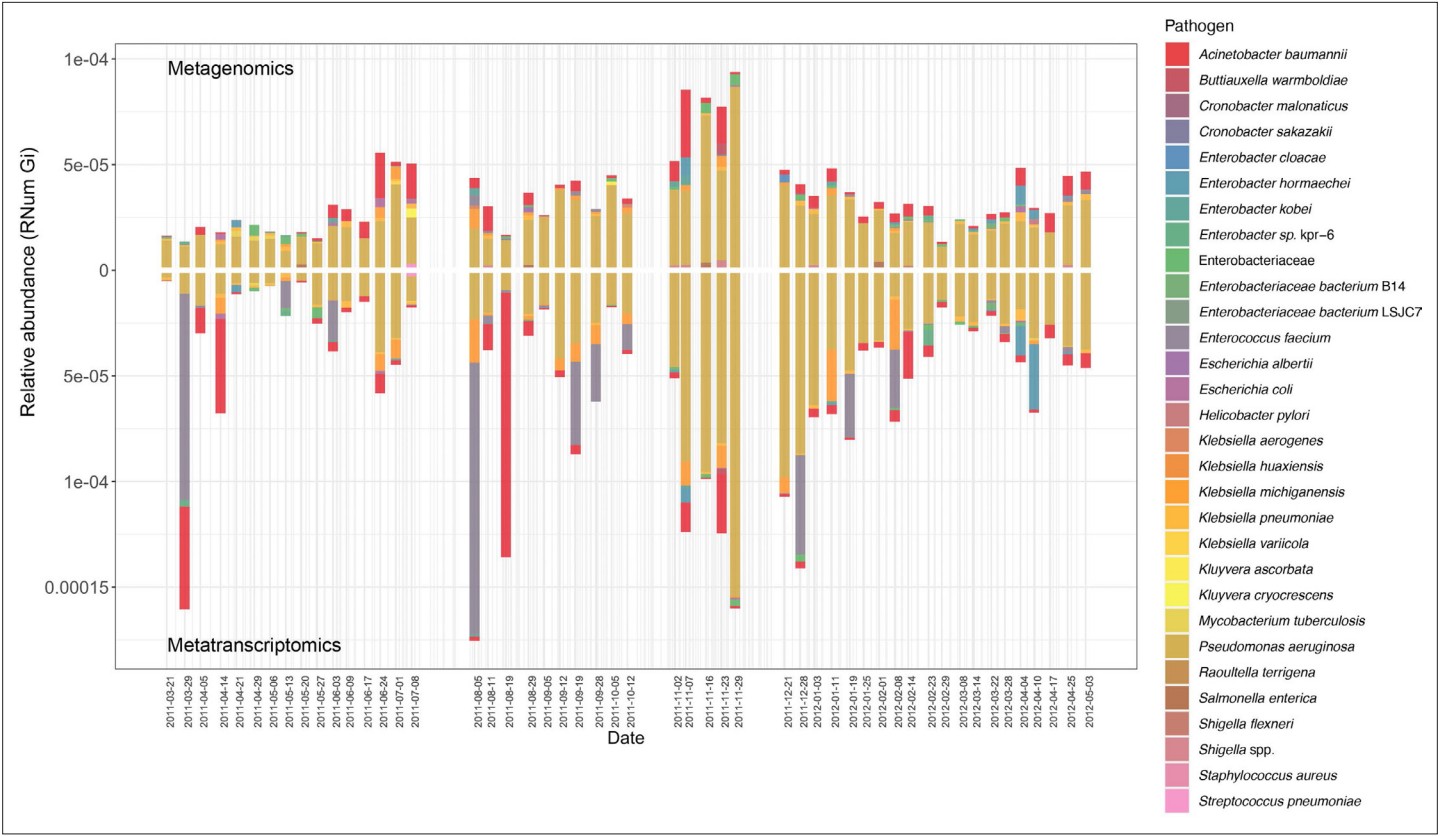

**Figure 3.** Assessment of AMR associated with clinical pathogens. ARG relative abundances encoded and expressed by clinical pathogens over time within the BWWTP, with colors corresponding to the identified pathogens.

well as plasmids (*Figure 4d*). For instance, the abundance in aminoglycoside and sulfonamide resistance, which was encoded primarily by plasmids (*Figure 5—figure supplement 1a*), fluctuated widely over time in both phages and plasmids (*Figure 4c*). Additionally, plasmid-mediated sulfonamide resistance was reduced at 23-11-2011, followed by its highest abundance a week later (29-11-2011), while subsequently again decreasing. Similarly, in line with the above observations, fosfomycin and peptide resistance genes, while segregating within phages, demonstrated significant fluctuations over time (*Figure 4d*). In addition to the metagenome, we also contextualized the localization of the expressed ARGs within MGEs based on the metatranscriptomic information. Specifically, we found that plasmids demonstrated a significantly increased expression of aminoglycoside along with bacitracin and sulfonamide resistance genes, while the expression of glycopeptide, mupirocin and peptide resistance genes were primarily enriched in phages (*Figure 5a*). These observations pertaining to plasmid-mediated AMR were in line with the metagenomic findings (*Figure 4b*). Only peptide resistance was observed to be expressed via phages in contrast to the differential enrichment of fosfomycin resistance observable in the metagenomic data.

## Taxonomic affiliations of MGE-derived resistance genes

When assessing the differential contributions of MGEs to AMR, we found congruency between plasmids and phages to the AMR categories and taxonomic affiliations (*Figure 5b*). For example, in the metagenomic data MGEs (phage and plasmid) were predominantly associated with the same AMR category and subsequently the same taxa. However, some exceptions were observed with specific taxa associated with AMR either through plasmids or phages. For instance, MLS resistance in Bacteroidales and Nostocales was mediated solely through plasmids, whereas the same resistance category was mediated by phage in Bifidobacteriales, indicating segregation of AMR between taxa and MGEs.

As most bacteria harbor MGEs, we queried whether the MGE-mediated AMR categories were linked to the abundance of some of the earlier reported taxa. Interestingly, we found that peptide

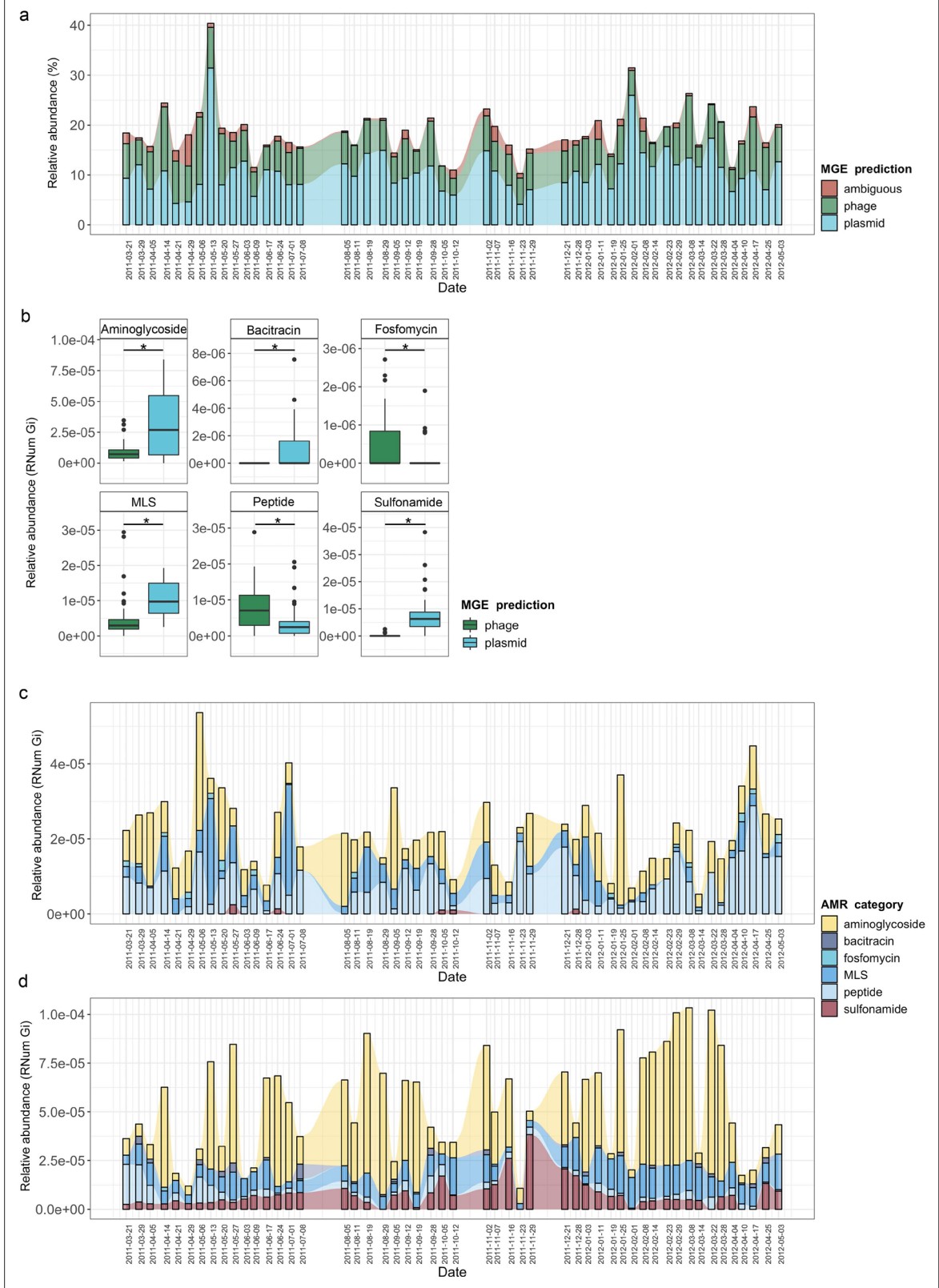

**Figure 4.** MGE-derived AMR within the BWWTP resistome. (**a**) Overall relative abundance of MGEs encoding ARGs. Contribution of plasmids to AMR (average of 10.8% of all ARGs) was found significantly increased compared to phages (average of 6.8% of all ARGs) (*adj.p* <0.05, One-way ANOVA). Colors depict the different MGE predictions (phage, plasmid, ambiguous) (**b**) Boxplots depicting significant (*adj.p* < 0.05, Two-way ANOVA) differential abundances of ARGs encoded by plasmids (blue) vs phages (green). (**c**) Relative abundance of the six significantly different AMR categories encoded

*Figure 4 continued*

on phages over time, with colors corresponding to AMR categories. (**d**) Relative abundance of the six significantly different AMR categories encoded on plasmids over time, with colors corresponding to AMR categories.

The online version of this article includes the following figure supplement(s) for figure 4:

**Figure supplement 1.** Partitioning of MGEs through AMR.

**Figure supplement 2.** Chromosomal-derived AMR.

resistance encoded by *M. parvicella* was solely associated with phages, while aminoglycoside resistance was primarily correlated with plasmids (*Figure 5—figure supplement 1b*). Other highly abundant taxa such as *Pseudomonas* and *Comamonas* (*Figure 5—figure supplement 1c-d*), on the other hand, were correlated with sulfonamide resistance in addition to aminoglycoside resistance encoded on plasmids (*Figure 5b*). This was further reflected within the metatranscriptome data where in taxa such as Acidimicrobiales the expression levels of aminoglycoside resistance were solely associated with plasmids (*Figure 5—figure supplement 2a*). Additionally, in the Burkholderiales family, peptide resistance was found to be expressed through phages (*Figure 5—figure supplement 2b*).

We also found a clear segregation of the mobilome with respect to individual pathogens in the metagenome. Interestingly, plasmids were exclusively associated with AMR in six out of the fourteen relevant taxa (*Figure 5c*). These included *Streptococcus pneumoniae*, *Staphylococcus aureus*, *Shigella flexneri*, *Klebsiella pneumoniae*, *Enterobacter kobei,* and *Enterobacter hormaechei*. Furthermore, the plasmids were also associated with conferring peptide, multidrug, MLS, beta-lactam, fluoroquinolone, bacitracin, aminoglycoside, aminoglycoside:aminocumarin and sulfonamide resistance. Phages were exclusively associated with glycopeptide and aminoglycoside resistance *in Salmonella enterica*. Overall, our results revealed for the first time the key segregation patterns of AMR via the mobilome in taxa that are of relevance to human health and disease. Moreover, substantiating the metagenomic data, the pathogenic bacteria *S. pneumoniae*, *S. aureus*, *K. pneumoniae*, *E. kobei,* and *E. hormaeche* were found to express ARGs solely associated with plasmids (*Figure 5c*). Collectively, these findings indicate an imminent threat to global health due to the potential dissemination of resistant pathogens across reservoirs.

## Metaproteomic validation of AMR abundance and expression

In order to expand our findings with the expression (metatranscriptomic) analyses on the BWWTP, we further used the corresponding metaproteomic data to offer complementary information at the protein level. Similar to the metagenome data we found protein expression linked to aminoglycoside, beta-lactam and multidrug resistance, over time within the BWWTP (*Figure 6—figure supplement 1*). Proteins, especially those linked to multidrug resistance were found to increase over time.

To further unravel AMR expression and assess its stability across time, we estimated the normalized protein index (NPI) per gene, as discussed in the Methods, by integrating all of the multi-omic data. The estimated NPI demonstrated stable levels of aminoglycoside and multidrug resistance within the BWWTP (*Figure 6a*). Specifically, proteins conferring multidrug resistance were found to increase over time, which is in line with the gene- and expression-level observations. Furthermore, we contextualized the normalized proteins conferring AMR to their localization on MGEs. We identified five resistance categories, that is aminoglycoside, beta-lactam, sulfonamide, multidrug, and tetracycline resistance, to be expressed through MGEs (*Figure 6b*). Of these categories we found that aminoglycoside resistance, in concordance with the gene and expression levels, was more strongly associated with plasmids than phages (*adj.p* <0.05, Two-way ANOVA). We further found that the MGE-mediated AMR categories were associated with specific microbial taxa. with plasmid-mediated aminoglycoside resistance found to be strongly associated with the previously mentioned *M. parvicella* (*Figure 6b*). On the other hand, we did not identify any peptides associated with the ESKAPEE pathogens via metaproteomics.

## Discussion

The surveillance of wastewaters for the identification of microbial molecular factors is a critical tool for identifying potential pathogens. This has been highlighted recently with the tracking of SARS-CoV-2

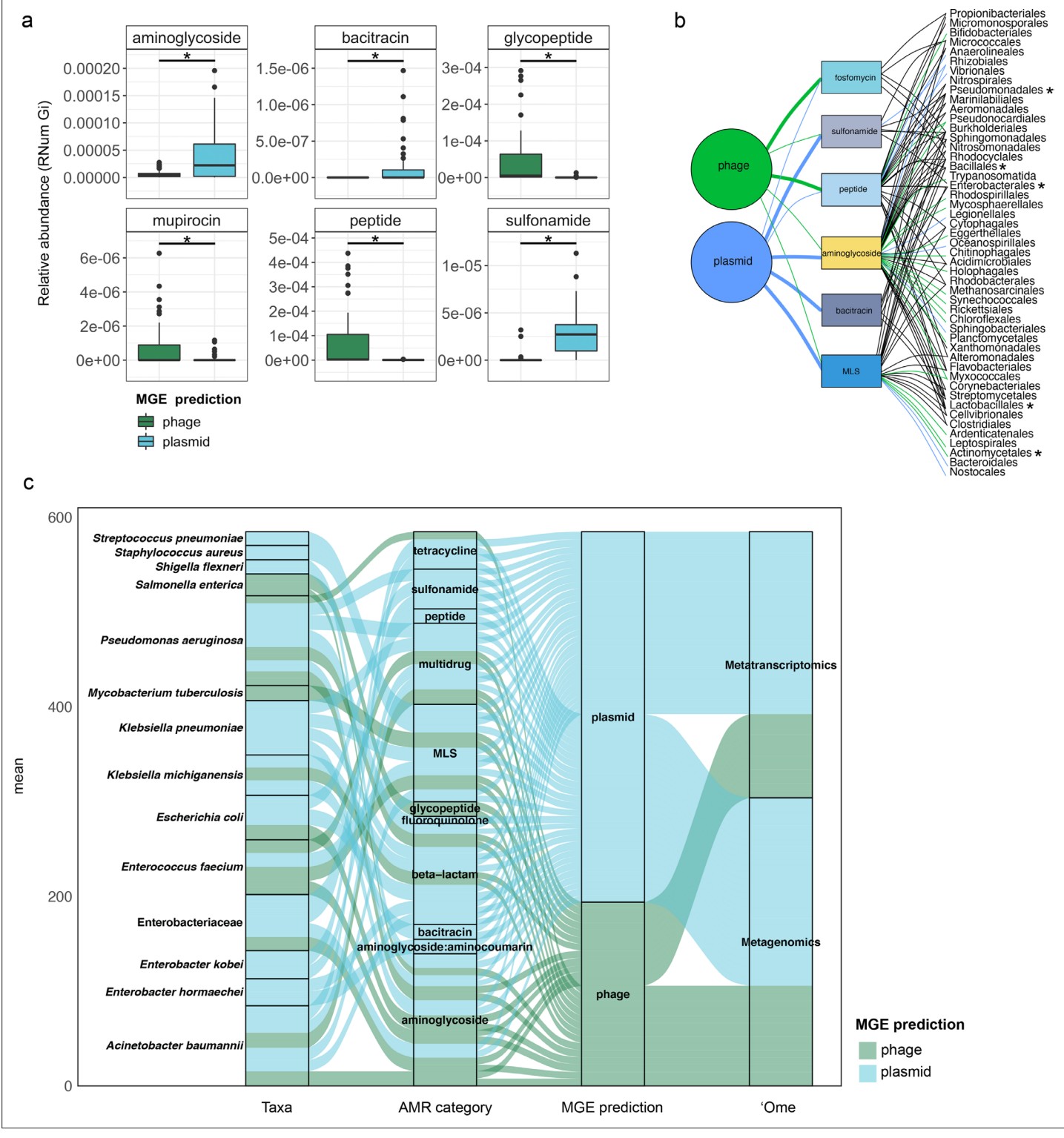

**Figure 5.** Taxonomic affiliations of MGE-derived resistance genes. (**a**) Boxplot depicting significant differential abundance (n=51 per group, *adj.p* <0.05, Two-way ANOVA) of ARGs expressed in plasmids vs phages. (**b**) Tripartite network assessing the association of MGE-derived ARGs with the microbial taxa. Thickness of the lines representing potential niche-partitioning of the AMR category to one MGE over the other. Color of the line representing which MGE the AMR is linked to: green (phage), blue (plasmid) or black (both phage and plasmid). Asterisk denominates taxonomic orders which include known clinical pathogens. (**c**) Alluvial plot depicting the mean abundance (log10) of MGE-derived ARGs encoded (metagenome) and/or expressed (metagenome) by clinical pathogens. Colors of all panels correspond to the MGEs and AMR categories.

*Figure 5 continued on next page*

*Figure 5 continued*

The online version of this article includes the following figure supplement(s) for figure 5:

**Figure supplement 1.** Differential ARG abundance in MGEs.

**Figure supplement 2.** Expression of AMR categories in MGEs.

within wastewater treatment plants to assess viral prevalence and load within a given community (*Westhaus et al., 2021*). Such approaches have also been employed for screening for antimicrobial resistance at a population level (*Kwak et al., 2015*; *Reinthaler et al., 2013*). So far, several studies (*Calero-Cáceres et al., 2014*; *Hendriksen et al., 2019*; *Parsley et al., 2010*; *Szczepanowski et al., 2009*) have characterized the proliferation of ARGs and antibiotic resistant bacteria in BWWTP. *Szczepanowski et al., 2009* identified 140 clinically relevant plasmid-derived ARGs in a BWWTP

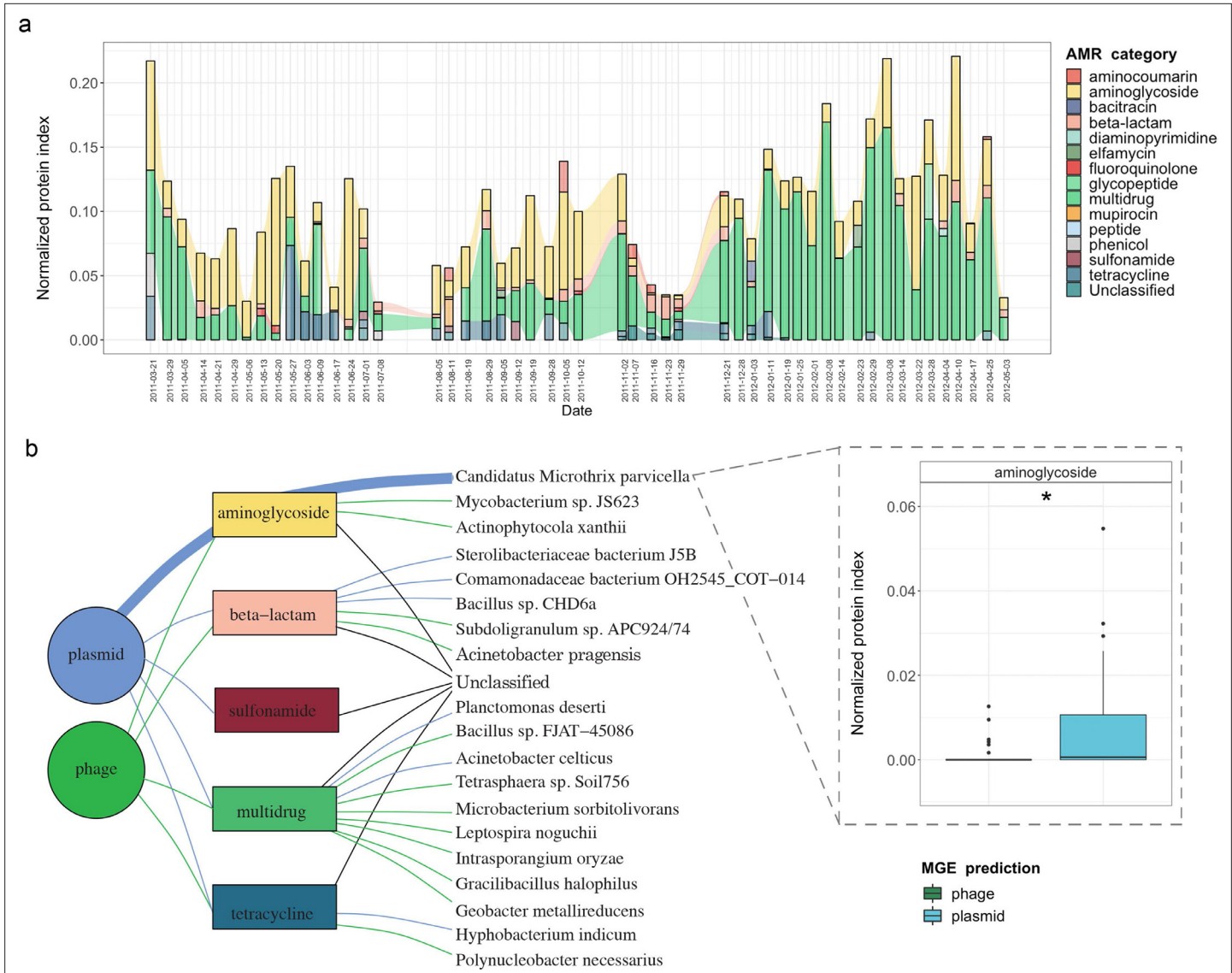

**Figure 6.** Integrative multi-omic assessment of AMR. (**a**) Metagenomic and metatranscriptomic normalized protein levels linked to AMR within the WTTP over time. (**b**) Tripartite network assessing the normalized protein levels derived from MGEs and associated taxa. Boxplots depicting significant differential (n=51 per group, *adj.p <0.05*, Two-way ANOVA) abundance of aminoglycoside resistance in plasmid versus phage in *Candidatus* Microthrix parvicella as well as overall. Colors of all panels correspond to the MGEs and AMR categories.

The online version of this article includes the following figure supplement(s) for figure 6:

**Figure supplement 1.** AMR protein abundances.

metagenome while *Parsley et al., 2010* characterized ARGs from bacterial chromosomes, plasmids and in viral metagenomes found in a BWWTP. Further studies have shown that conventional BWWTP processes at best only partially remove ARGs from the effluent and may find their way into the urban water cycle (*Hiller et al., 2019*; *Proia et al., 2018*; *Rodriguez-Mozaz et al., 2015*). Wastewater treatment plants, therefore, are crucial reservoirs of AMR, whose monitoring may allow for early-detection of AMR within the human population feeding into the system. Here, we leveraged a systematic and longitudinal sampling scheme from a BWWTP to identify diverse AMR categories prevalent within the BWWTP microbial community. In line with the studies by *Szczepanowski et al., 2009* and *Parsley et al., 2010*, we found up to 29 AMR categories with several ARGs within the BWWTP. More importantly, and unlike the previous studies, we linked the identified ARGs to clinically-relevant ESKAPEE pathogens, which represent a growing global threat to human health.

In our BWWTP samples, we identified a core group of 15 AMR categories that were ubiquitous at all timepoints. In line with the above-mentioned reports, the observed core resistance categories may reflect their abundance in the surrounding human population (*Aarestrup and Woolhouse, 2020*). This has previously been reported by *Pärnänen et al., 2019*, *Su et al., 2017* and *Hendriksen et al., 2019* where they showed that BWWTP AMR profiles correlate with clinical antibiotic usage as well as other socio-economic and environmental factors. Furthermore, bacteria are known to have innate defense mechanisms against inhibitory secondary metabolites from other taxa (*Frost et al., 2018*). Therefore, one must be cognizant of thprace phenomenon that the observed core group of AMR categories may also be a proxy for the abundance of specific resistant bacteria. Despite this observation, it is plausible that both anthropogenic and microbial sources for AMR play a role in the observed resistance categories within the BWWTP. For instance, we found that several AMR categories, including ancillary

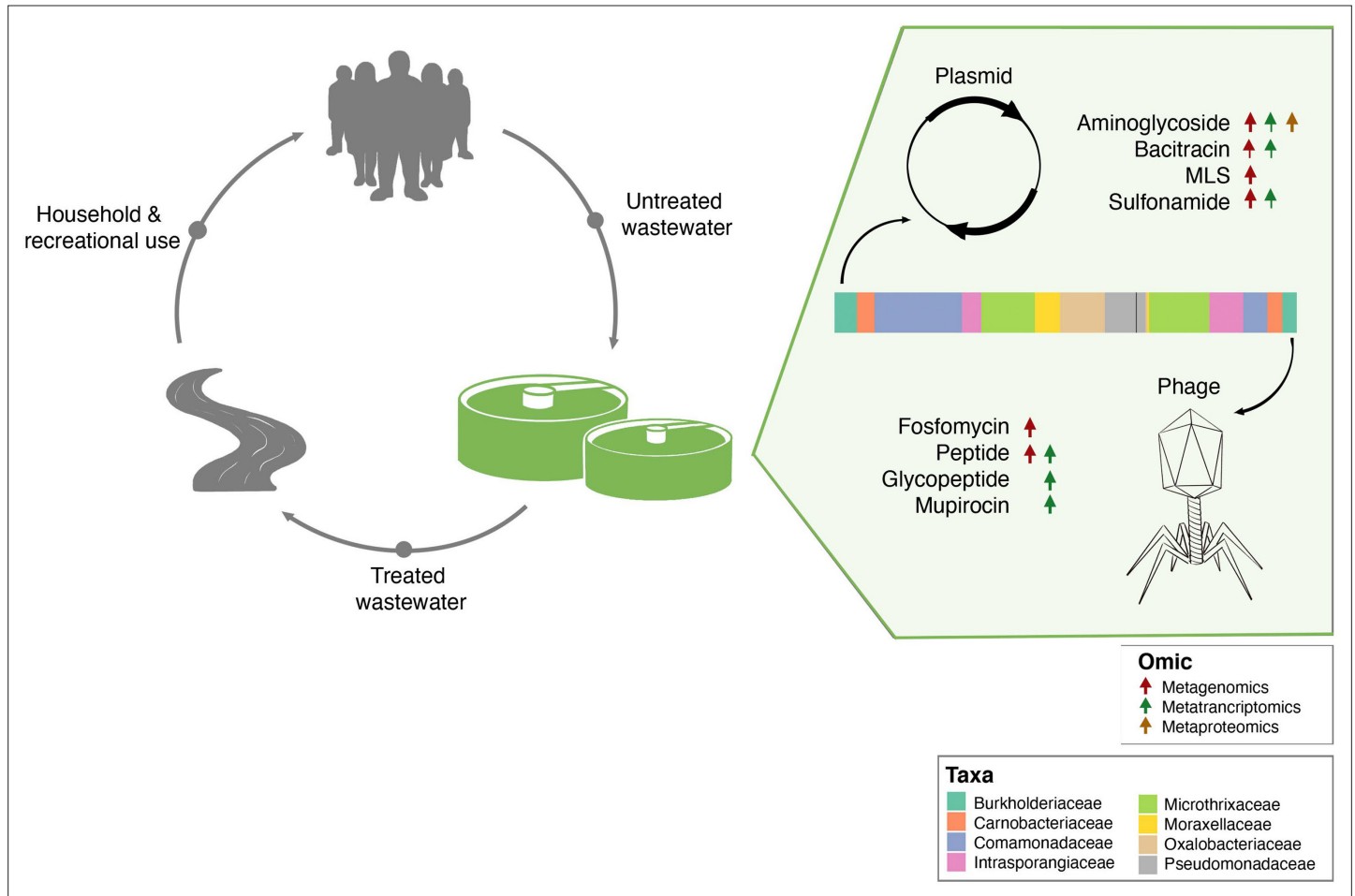

**Figure 7.** Separation of MGE-derived AMR within the BWWTP. A graphical summary highlighting AMR categories found significantly increased in phage versus plasmid in all three omes.

(prevalent, moderate, and rare) groups, were associated with *M. parvicella* within the BWWTP. While the mechanisms underlying the abundance of this taxon are numerous, the increased abundance of ARGs are likely to contribute to its fitness advantage in this metal- and antibiotic-rich environment (*Baquero et al., 2008*). Additionally, similar to the findings by *Munck et al., 2015*, we found a wide range of bacteria associated with AMR categories including Acidimicrobiales, Burkholderiales and Rhodocyclales. On the other hand, we report that taxa, including ESKAPEE pathogens, belonging to 25 bacterial orders were associated with 29 AMR categories, compared to the eight bacterial orders reported previously.

It is important to note that the mobilome plays a critical role in the dissemination of AMR within microbial communities. AMR from resistant bacteria within the BWWTP can quickly disseminate within the BWWTP (*Che et al., 2019*; *Fouz et al., 2020*), including transmission from pathogenic to commensal species (*Blake et al., 2003*; *Brinkac et al., 2017*). As a result, mediated through HGT, the BWWTP becomes a hotspot for resistant bacteria, which are then released back into the receiving environment (*Turolla et al., 2018*), and eventually the human population (*Fouz et al., 2020*; *Newton and McClary, 2019*). Therefore, to limit the dissemination of AMR, it is important to understand the role of MGEs within the BWWTP. Our comprehensive analyses identified the differential contributions of AMR transmission mediated via phage and plasmid (*Figure 7*). Specifically, we identified clear segregation of aminoglycoside, bacitracin, MLS and sulfonamide resistance categories with plasmids, while fosfomycin and peptide resistance were increasingly encoded and conferred via phages. While the association between these AMR categories and plasmids (*Dubnau et al., 1981*; *Galimand et al., 2003*; *Han et al., 2015*; *Razavi et al., 2017*) or phages (*Torres-Barceló, 2018*) are in line with previously reported results, differential analysis between MGEs has not been previously reported and has not been performed on multi-omic levels. In this study we report for the first time the systematic and extensive comparison of AMR encoded and expressed by phages versus plasmids. Our results indicating the segregation of ARGs within the ESKAPEE taxa via the MGEs further provide insights into potential modes of AMR transmission among pathogens. Although one cannot exclude the possibility of transmission of the above-mentioned ARGs via other MGEs, identifying potential segregation of MGEs in the transmission of ARGs brings us one step closer to identifying specific transmission paths and limiting the spread of AMR. For example, some studies have reported plasmid 'curing', the process by which plasmids are lost from bacterial populations, as a strategy against dissemination of AMR (*Bouanchaud and Chabbert, 1971*; *Vrancianu et al., 2020*). As described by *Buckner et al., 2018* plasmid curing, as well as other anti-plasmid strategies, could both reduce AMR prevalence, and (re-)sensitize bacteria to antibiotics (*Buckner et al., 2018*).

Although several methods and tools exist for the identification of MGEs, the linkage to their respective bacterial hosts still remains a challenge. Our method presented here involves taxonomic annotations of reconstructed genomic information as a mean of linking bacteria to their corresponding MGEs using stringent filtering/identification criteria. An alternative approach may involve targeted sequencing of plasmids of interest and their respective hosts using the methodology established by *Li et al., 2018*. However, to use such targeted sequencing approaches one requires a priori information on which plasmids to focus on in contrast to our method which is agnostic to such prior information. Furthermore, Hi-C-based methods (*Lieberman-Aiden et al., 2009*; *Press et al., 2017*) capture inter-chromosomal junctions, such as plasmid-genome interactions. However, compared to metagenomic approaches, these methods require different and extensive processing of the samples prior to sequencing to induce covalent linkages among DNA molecules (*Lieberman-Aiden et al., 2009*; *Press et al., 2017*), therefore precluding the use of this method for large-scale characterization efforts. Combining these strategies in future studies with AMR categorization according to taxonomic affiliation as well as linkage to specific MGEs will provide novel strategies for characterizing and subsequently affecting MGE-mediated AMR.

By complementing the metagenomic analyses, metatranscriptomics conferred essential information regarding gene expression within the resistome. For instance, when comparing AMR expression levels of aminoglycoside, bacitracin, and sulfonamide mediated via MGEs, it is noticeable that expression levels in plasmids mirror the genomic content, that isthey exhibited higher levels of expression when compared to phage. On the other hand, glycopeptide and mupirocin resistance genes which were highly expressed in phages were not reflected within the metagenomic data. Additionally, we found the *YojI* resistance gene to be more highly expressed than any other ARGs. To facilitate resistance

against the peptide antibiotic microcin J25, the outer membrane protein, TolC, in combination with *YojI* is required to export the antibiotic out of the cell (*Delgado et al., 2005*). Microcin J25 belongs to the group of ribosomally synthesized and post-translationally modified peptides (RiPPs) and has antimicrobial activity against pathogenic genera such as *Salmonella* spp. and *Shigella* spp. (*Naimi et al., 2018*). Interestingly, it has only recently been proposed as a treatment option against *Salmonella enterica and* has been discussed in recent years as a potential novel antibiotic (*Ben Said et al., 2020*). Based on these results, by considering that BWWTPs may reflect both the presence of AMR within the human population as well as be a hotspot of dissemination and generation of new AMR, surveillance of BWWTPs must be emphasized when developing new antibiotics. Our findings collectively suggest that the differential capacity of MGEs to disseminate AMR, coupled with longitudinal and expression-level analyses are crucial for monitoring human health conditions. More importantly, we report for the first time that BWWTP monitoring for AMR may allow for early detection of resistance mechanisms previously undescribed in BWWTP.

Finally, we applied an integrated multi-omic analysis approach to improve our knowledge on the functional potential of AMR and simultaneously validate the abundance and expression findings of the ARGs. By normalizing the metaproteomic results with the normalized expression of genes, we were able to assess the stability of expressed AMR across time. We find that our methodology allows for an unbiased assessment of overall expression accounting for gene copy abundance and expression. These findings support the notion that the AMR genes may serve as sentinels or indicators of the presence of particular antimicrobial agents. However, it is plausible that we are only identifying the most abundant proteins and/or proteins that are more stable over time, and do not capture the entirety of the proteome profiles. Factors such as protein decay rates (*Cameron and Collins, 2014*) among others, may additionally influence this assessment. Irrespective of these observations, we identified segregation of AMR categories with respect to plasmids and phages.

Our findings also highlight the potential for identifying segregation of AMR via specific MGEs with an aim toward possible therapeutic and mitigation strategies via for example plasmid curing. Furthermore, we demonstrate that longitudinal analyses are required to survey AMR within BWWTPs due to the variations in the resistome across time. These shifts may either be representative of a shift within the human population itself, which in turn could be associated with the concurrent use of antibiotics at a given time, or competition within the microbial community. In any case, an independent or static analysis of the various time points may show an incomplete view of the BWWTP resistome, thus underlining the importance of our longitudinal resistome analyses. Overall, our findings suggest that BWWTPs are critical reservoirs of AMR, potentially allowing for early detection and monitoring of pathogens. In addition, BWWTP monitoring may allow detection of resistance mechanisms linked to the introduction of new antimicrobials. Finally, BWWTPs may serve as a model for understanding the segregation of MGEs through AMR.

## Methods

### Sampling and biomolecular extraction

From within the anoxic tank of the Schifflange municipal biological wastewater treatment plant (located in Esch-sur-Alzette, Luxembourg; 49° 30′ 48.29″ N; 6° 1′ 4.53″ E) individual floating sludge islets were sampled according to previous described protocols (*Herold et al., 2020*). Sampling was performed starting on 21-03-21 till 03-05-2012 in approximately one-week intervals resulting in a total of 51 samples. DNA, RNA and proteins were extracted from the samples in a sequential co-isolation procedure as previously described (*Roume et al., 2013*).

### Sequencing and data processing for metagenomics and metatranscriptomics

Paired-end libraries were generated for metagenomics with the AMPure XP/Size Select Buffer Protocol following a size selection step recommended by the standard protocol. Libraries for metatranscriptomics were prepared from RNA after washing stored extractions with ethanol and depletion of rRNAs with the Ribo-Zero Meta-Bacteria rRNA Removal Kit (Epicenter). Subsequently, the ScriptSeq v2 RNA-seq library preparation kit (Epicenter) was used for cDNA library preparation, followed by sequencing on an Illumina Genome Analyses IIx instrument with 100-bps paired-end protocol. Processing and

iterative co-assembly of metagenomic and metatranscriptomic reads was done using the Integrated Meta-omic Pipeline (*Narayanasamy et al., 2016*) (IMP v1.3; available at https://r3lab.uni.lu/web/imp/). For read processing, Illumina Truseq2 adapters were trimmed, and reads of human origin were filtered out, followed by a de novo assembly with MEGAHIT (*Li et al., 2015*) v1.0.6. For the assembly, both metagenomic and metatranscriptomic reads were co-assembled to increase contiguity of the assemblies and to improve read usage. Additional information regarding the read coverage and depth for each sample is available at https://git-r3lab.uni.lu/laura.denies/lao_scripts.

## Identification of antimicrobial resistance genes, mobile genetic elements and taxonomy

The assembled contigs from IMP were used as input for PathoFact (*de Nies et al., 2021*), for the prediction of antimicrobial resistance genes and MGEs. ARGs were further collapsed into their respective AMR categories, as identified by PathoFact in accordance with those provided by the Comprehensive Antibiotic Resistance Database (CARD) (*Alcock et al., 2020*). Furthermore, utilizing PathoFact, AMR genes were linked to predicted MGEs (i.e. plasmids and phages) to track transmission of AMR. By considering all different predictions of MGEs, a final classification was made by PathoFact based on the genomic contexts of the AMR genes encoded on plasmids, phages or the organismal chromosomes, including the classification of those that could not be resolved (ambiguous). The AMR genes

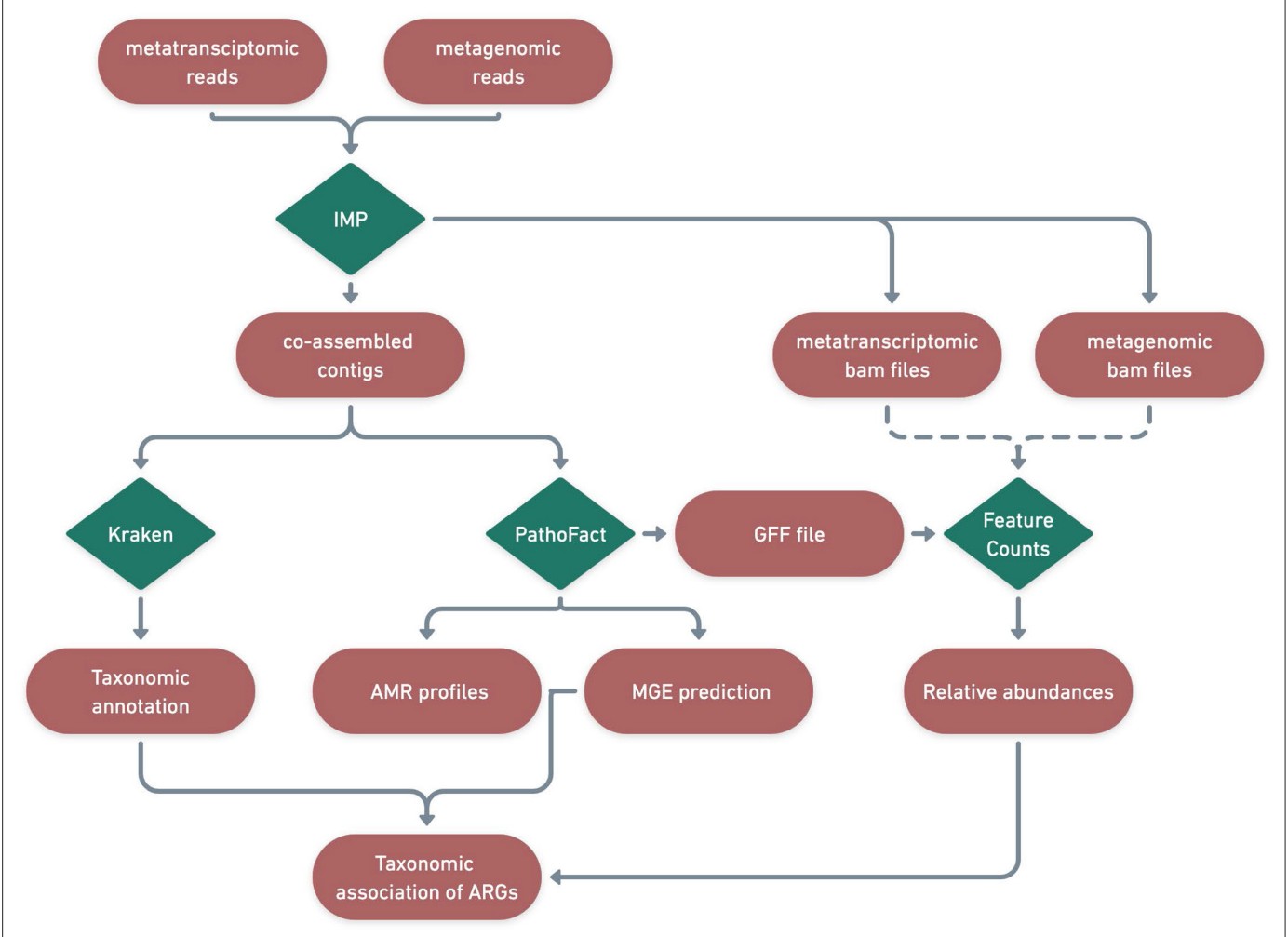

**Figure 8.** Identification of ARGs and contextualization of MGEs in relation to taxa. A schematic diagram depicting the bioinformatic workflow to identify ARGs and the subsequent contextualization of MGEs in relation to microbial taxa. The green features represent the different tools and pipelines used while the red features highlight the data used and generated in this process.

that could not be assigned to either the MGEs or bacterial chromosomes were subsequently referred to as unclassified genomic elements.

Identified ARGs and their categories were further linked to associated microbial taxonomies using the taxonomic classification system Kraken2 (*Wood et al., 2019*). Kraken2 was run on the contigs using the maxikraken2_1903_140 GB (March 2019, 140GB) (https://lomanlab.github.io/mockcommunity/mc_databases.html) database (*Wood et al., 2019*). To ensure confidence in taxonomic classification, stringent cutoffs based on a minimum 70% identity of the contig with the database, across 90% of the contig length, was used (*Wood et al., 2019*). Since the same contig-based assembly file was used as input for PathoFact as well as for Kraken2, the prediction of antimicrobial resistance genes and mobile genetic elements as well as functional- and taxonomic- annotations were linked based on the contig they were encoded on (*Figure 8*).

Though several methods and tools exist for the identification of plasmids and/or phages, tools identifying the respective bacterial hosts for MGEs are rare. The hosts for phages can be determined based on classification and linking of CRISPR and spacer sequences between bacterial and phage sequences respectively (*Bland et al., 2007*; *Martínez Arbas et al., 2021*). Plasmid hosts, on the other hand, due to the competence of plasmids (including self-replication, etc.), are difficult to identify (*Suzuki et al., 2010*) or can only be classified against existing databases (*Aytan-Aktug et al., 2022*).

Therefore, to streamline this process, we used taxonomic associations as a means of linking bacteria with their corresponding MGEs using the stringent filtering/identification criteria. Based on the already established stringent cutoffs for plasmid/phage predictions, the original MGE predictions from PathoFact were used without any additional processing and the taxonomic classifications from Kraken2 were used to assign putative hosts. Finally, to obtain the gene copy number and transcriptome expression levels, both metagenomic and metatranscriptomic reads were independently mapped to the corresponding assembly files per sample. The raw read counts per contig as well as per individual ORF, as given by PathoFact, were determined with the featureCounts option. The relative abundance of the ARGs was calculated using the RNum_Gi method previously described by *Hu et al., 2013*.

## Metaproteomics and data analyses

Raw mass spectrometry files were converted to MGF format using MSconvert (*Chambers et al., 2012*) with default parameters. The metaproteomic searches were performed using SearchGUI / Peptide-Shaker (*Vaudel et al., 2015*) for each time point. To generate the databases, each predicted protein sequence file was concatenated with the cRAP database of contaminants (*common Repository of Adventitious Proteins*, v 2012.01.01; The Global Proteome Machine) and with the human UniProtKB Reference Proteome (*Consortium, 2021*). In addition, inverted sequences of all protein entries were concatenated to the databases for the estimation of false discovery rates (FDRs). The search was performed using SearchGUI-3.3.20 (*Barsnes and Vaudel, 2018*) with the X!Tandem (*Langella et al., 2017*), MS-GF+ (*Kim and Pevzner, 2014*) and Comet (*Eng et al., 2013*) search engines using the following parameters: Trypsin was used as the digestion enzyme and a maximum of two missed cleavage sites was allowed. The tolerance levels for identification were 10 ppm for MS1 and 15ppm for MS2. Carbamidomethylation of cysteine residues was set as a fixed modification and oxidation of methionines was allowed as variable modification. Peptides with a length between 7 and 60 amino acids and with a charge state composed between +2 and+4 were considered for identification. The results from SearchGUI were merged using PeptideShaker-1.16.45 (*Vaudel et al., 2015*) and all identifications were filtered in order to achieve a peptide and protein FDR of 1%.

Each predicted protein sequence corresponded to the predicted ORFs generated by the Prodigal (version 2.6.3) predictions included in PathoFact. As such predicted protein sequences matched the ARG annotation of the ORFs as provided by PathoFact.

## Multi-omic data integration

To further improve upon the understanding of the AMR expression and assess its stability across time, we estimated the normalized protein index (NPI) per gene, by integrating the multi-omic data. To estimate the NPI, we first normalized the metaT abundance based on per gene copy numbers obtained via the metagenomic abundance:

$$NPI = \frac{N_{metaproteome}}{N_{metatranscriptome} / N_{metagenome}}$$

This, the normalized expression of genes, yields the per copy expression of ARGs within each AMR category. Subsequently, the normalized expression was used to standardize the metaP abundances for those genes where the necessary data was available.

## MGE partition assessment

To assess the segregation of MGEs through AMR we determined niche regions and overlap using the *nicheROVER* R package (*Swanson et al., 2015*). *nicheROVER* uses Bayesian methods to calculate niche regions and pairwise niche overlap using multidimensional niche indicator data (i.e. stable isotopes, environmental variables). As such, using AMR as the indicator data, we extended the application of *nicheRover* to calculate the probability for the size of the niche area of one MGE inside that of the other, and vice versa. We calculated the segregation size estimate for each MGE and additionally generated the posterior distributions of µ (population mean) for each AMR category in all omics. We further computed the niche overlap estimates between MGEs with a 95% confidence interval over 10,000 iterations.

## Data analysis

Figures for the study including visualizations derived from the taxonomic and functional analyses were created using version 3.6 of the R statistical software package (*R Development Core Team, 2013*). A paired two-way ANOVA (Analysis of Variance, accounting for MGEs and AMR, or taxa and AMR) within the *nlme* package was used for identifying statistically significant differences. Tripartite and the association networks were generated using the *SpiecEasi* (*Kurtz et al., 2015*) R package where a weighted adjacency matrix was generated using the Meinhausen and Buhlmann (*mb*) algorithm, with a nlambda of 40, and lambda minimum ratio at 0.001. The analyses were bootsrapped with n=999 to avoid overfitting, autocorrelations and false network associations. The network was further refined, selecting for positive edges, with a degree greater than the mean-degree of the initial network. Additionally, edges with a correlation value greater than 0.6 were retained where the associated significance was below an adjusted p-value <0.05. The association networks were based on two types of edges, that is, co-occurences between taxa based on ARG abundances and ARG-taxon edges based on ARGs encoded on specific taxa. The *igraph* (*Csardi and Nepusz, 2006*) package was used in R to render the graphics for the network. All code for visualization and analysis is available at: https://git-r3lab.uni.lu/laura.denies/lao_scripts.

## Acknowledgements

We are thankful for the assistance of Audrey Frachet Bour, Lea Grandmougin, Janine Habier, Laura Lebrun (LCSB) for laboratory support. We acknowledge the valuable input from Rashi Halder at the LCSB Sequencing Platform with respect to library preparation. The computational analyses were performed at the HPC facilities at the University of Luxembourg (*Varrette et al., 2014*).

## Additional information

### Funding

| Funder | Grant reference number | Author |
| --- | --- | --- |
| Fonds National de la Recherche Luxembourg | CORE/13684739 | Paul Wilmes |
| European Research Council | ERC-CoG 863664 | Paul Wilmes |
| Fonds National de la Recherche Luxembourg | PRIDE/11823097 | Paul Wilmes Laura de Nies |

| Funder | Grant reference number | Author |
|---|---|---|
| Schweizerischer Nationalfonds zur Förderung der Wissenschaftlichen Forschung | CRSII5_180241 | Susheel Bhanu Busi |

The funders had no role in study design, data collection and interpretation, or the decision to submit the work for publication.

## Author contributions

Laura de Nies, Conceptualization, Resources, Data curation, Software, Formal analysis, Validation, Investigation, Visualization, Methodology, Writing – original draft, Writing – review and editing; Susheel Bhanu Busi, Resources, Data curation, Formal analysis, Investigation, Visualization, Writing – original draft, Writing – review and editing; Benoit Josef Kunath, Formal analysis; Patrick May, Data curation, Software, Writing – original draft, Writing – review and editing; Paul Wilmes, Conceptualization, Resources, Software, Supervision, Funding acquisition, Validation, Writing – original draft, Project administration, Writing – review and editing

## Author ORCIDs

Laura de Nies (iD) http://orcid.org/0000-0002-6483-7489
Susheel Bhanu Busi (iD) http://orcid.org/0000-0001-7559-3400
Benoit Josef Kunath (iD) http://orcid.org/0000-0002-3356-8562
Patrick May (iD) http://orcid.org/0000-0001-8698-3770
Paul Wilmes (iD) http://orcid.org/0000-0002-6478-2924

## Decision letter and Author response

Decision letter https://doi.org/10.7554/eLife.81196.sa1
Author response https://doi.org/10.7554/eLife.81196.sa2

# Additional files

## Supplementary files

• MDAR checklist

## Data availability

The genomic FASTQ files used in this work (previously published) are publicly available at NCBI BioProject PRJNA230567. Metaproteomic data (previously published) are publicly available at the PRIDE database under accession number PXD013655. The open-source tools and algorithms used for the data analyses are reported in the Methods section, including relevant flags used for the various tools. Additionally, custom code for further analysis and generation of the figures can be found at: https://git-r3lab.uni.lu/laura.denies/lao_scripts.

The following previously published datasets were used:

| Author(s) | Year | Dataset title | Dataset URL | Database and Identifier |
|---|---|---|---|---|
| Arbas S, Narayanasamy S, Herold M, Lebrun LA, Hoopmann MR, Li S, Kunath BJ, Hicks ND | 2021 | Systems Biology of Lipid Accumulating Organisms | https://www.ncbi.nlm.nih.gov/bioproject/?term=PRJNA230567 | NCBI BioProject, PRJNA230567 |

*Continued on next page*

*Continued*

| Author(s) | Year | Dataset title | Dataset URL | Database and Identifier |
|---|---|---|---|---|
| Herold M, Martínez Arbas S, Narayanasamy S, Sheik AR, Kleine-Borgmann LAK, Lebrun LA, Kunath BJ, Roume H, Bessarab I, Williams RBH, Gillece JD, Schupp JM, Keim PS, Jäger C, Hoopmann MR, Moritz RL, Ye Y, Li S, Tang H, Heintz-Buschart A, May P, Muller EEL, Laczny CC, Wilmes P | 2020 | Systems Biology of Lipid Accumulating Organisms | https://www.ebi.ac.uk/pride/archive/projects/PXD013655 | PRIDE, PXD013655 |

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
