## [Editor Report]

This paper reports important results regarding the presence and potential dissemination of antibiotic resistance genes in wastewaters by convincingly combining analysis of gene abundance, expression, and association with mobile genetic elements and bacterial taxa. Via systematic evaluation and implementation of multiple tools, the authors provide a valuable approach for monitoring antibiotic resistance genes in the environment and assessing their dispersal and possible risks to human health.

---

## [Decision Letter]

**Decision letter after peer review:**

[Editors’ note: the authors submitted for reconsideration following the decision after peer review. What follows is the decision letter after the first round of review.]

Thank you for submitting the paper "Mobilome-driven segregation of the resistome in biological wastewater treatment" for consideration by *eLife*. Your article has been reviewed by 2 peer reviewers, including María Mercedes Zambrano as the Reviewing Editor and Reviewer #1, and the evaluation has been overseen by a Senior Editor.

Comments to the Authors:

We are sorry to say that, after consultation with the reviewers, we have decided that this work will not be considered further for publication by *eLife*.

Specifically, more methodological details and rigorous statistical analyses are needed in several places, as well as more thorough and clearer information to fully assess the validity of the results and the subsequent claims.

*Reviewer #1 (Recommendations for the authors):*

This work aims to provide a broad view of antimicrobial resistance determinants in a wastewater treatment plant by conducting a multi-omics approach on samples taken over 1.5 years. The use of several omics approaches and the distinction of markers associated with either phages or plasmids in the system represent strengths of the study. However, the lack of clarity and methodological details weaken the interpretation and impact of the results.

1. While the multiple omics approach provides valuable insight, it is not entirely clear how expression data adds to what is already known regarding ARGs in the environment and how this expression information sheds new light on the resistome or can be "crucial for monitoring human health conditions" (L412). In this respect it would have been good to see if their findings on ARGs segregating between phages and plasmids correlate with what is known about resistances, at least in the relevant taxa (eg, L 288-9).

2. Since the results are based on assembled data (L 158), the authors should provide additional and sufficient information regarding these assemblies and how they can be confident of taxa identification to the species level (such as the ESKAPEE group) and the linkage of ARGs to particular microorganisms. This is important, given that much of the analysis and ensuing conclusions are based on linkage of ARGs and MGEs to particular taxa. It would also be important to know how they managed to precisely map expression data to these MGEs and not to chromosomal loci and if perhaps this could also explain some of the discrepancies observed (eg, L397-8).

3. It seems that about 18% of all ARGs are associated with MGEs, yet no additional mention is made of the remaining genes, which presumably are chromosomal. Please clarify this in the document and also indicate how this could affect the transcriptomic and proteomic analysis.

4. It is intriguing that they observe high levels of peptide resistance in their metatranscriptome (L146) associated with resistance in *E. coli*, which is not a major contributor in the community and the gene was apparently not abundant in their metagenomic analysis. This discrepancy is not fully addressed and is relevant because it can also reflect biases in the approach or conditions of the sites being analyzed.

5. It seems that the majority of markers are associated with abundant taxa, which is to be expected, but it is not clear if this is the reason why the authors hypothesize that the majority of ARGs would be in the major taxon M. parvicella (L169 and L175) or why these ARGs would confer fitness advantage (L184), unless of course this correlates with antibiotics present, which was not determined.

6. The document can also benefit from revision of the text, particularly the discussion, to clarify some claims and make the document easier to follow. There are some statements that do not accurately reflect either the scope of the work or the results obtained. Some of these include:

L79 – why would ARGs and sub-inhibitory antibiotic concentrations (which should not be lethal) promote HGT?

L 96 – They do not really shed light on "the evolution" and at best suggest dissemination differences.

L102 – The analysis suggests but does not necessarily "demonstrate" that MGEs are important drivers of AMR dissemination. In addition, it does not necessarily follow that "assessing the activity of the ARGs" can lead to understanding the underlying mechanisms.

L186 – why population sizes? Did you actually look at association with the size?

L192 – Is this figure 3a?

On L202 it says that they "assessed the acquisition and dissemination of AMR". The data shows only associations, which is not necessarily acquisition and at best hints at modes of dissemination.

L292-94- the findings indicate but are not themselves a threat or have potential for dissemination. Please correct.

L296 – it does not seem that the proteomics validated transcriptomics data, but rather are more of a complementary analysis (L298). Likewise, on L416- It does not seem that proteomics data validated the abundance and expression findings.

L413 and 438 – Monitoring based on sequence analysis is based on comparisons using known data and therefore it would be difficult to detect undescribed or novel mechanisms.

*Reviewer #2 (Recommendations for the authors):*

In this paper, de Nies and colleagues reanalyse their previously released, ground-breaking multi-omics time series dataset collected from a wastewater treatment plant microbial community over a 14 month period (with 51 sampling days), with the aim of exploring the type and diversity of AMR-related genes present in this microbial community, the type of mobile genetic element they are associated with (plasmid or phage), their association with microbial groups and the nature of their temporal dynamics.

This work holds strong interest and utility for workers in the areas of antimicrobial resistance and One Health, and, more broadly, for our understanding the nature of the interconnections between human and natural ecosystems.

The starting point is the metagenome assemblies constructed from the collected nucleic acid sequence datasets, and from which contigs form the units of analysis. The analysis is organised around classifying AMR related genes within contig sequences, with particular reference to distinguishing between the type of mobile genetic element (MGE) that harbour them, namely either plasmids or bacteriophage/prophages (assumed to be incorporated into a host genome via transduction). While there are definitive strengths of using the contig-level analysis, I feel far more detail could be provided in the methods description to convince the reader of the validity of the analysis, particularly in relation to drawing associations between MGE sequences and host microbial groups. Exactly how this is done, and the relative strengths and weaknesses, are not fully described and this limits the extent to which the reader can evaluate the overall approach and findings.

At certain places in the manuscript, I feel there is an over-reliance of visual data analysis, and more rigorous statistical analysis should be used to test the hypotheses in question; this includes (1) the analysis summarised on line 222 concerning the proportion of ARG/s attributable to plasmids and phages, and (2) the analyses presented in Figure 2C. I note there is also an analysis examining ESKAPEE pathogens, however, there is no direct evidence that the genome "backbone" of these organisms are actually present.

Materials and methods

For an analysis of considerable complexity, the methods section needs more detail and clarity, as it is sometimes hard to understand how a specific result as been obtained.

Firstly, the metagenome and metatranscriptome reads from all n=51 samples are co-assembled (line 462-463 ), but it is not clearly stated whether the both sets of data are assembled together?, This appears to be the case, as the reader is given the impression there is a single set of contigs in play? (line 466).

Secondly, a critical part of the methodology relates to the annotation of MGE sequences, and the inference of association between MGE sequences and bacterial groups; however. My understanding is that PathoFact will predict whether a contig arises, if classifiable, from either a plasmid or a phage, or whether it derives from a chromosomal genome. The taxonomic annotations for all contig are then made with Kraken. One possibility is that a contig arises from a chromosomal genome and contains an AMR sequence, which seems straightforward, as cognate AMR sequence will be unambiguously associated with the Kraken taxonomic annotation. Whereas another possibility is that the contig arises from a non-chromosomal MGE context; but in that case, it is a little unclear how the taxonomic assignments are actually made (i.e. over lines 477-489); particularly the logic that supports the statement "…to track transmission of AMR between taxa". It is very unclear how these decisions are actually made, but it comes across as rather subjective e.g. "…considering all different predictions of MGEs…". Needless to say, these inferences are critical for the rest of the paper, so can the authors please describe the approach in more detail, including backfilling the underlying scientific logic? A schematic diagram would be useful showing how different sources of data are linked.

Results

Figure 1. What is the level of read coverage that underpins the relative abundance measures used? Can some short statistical summary to cover that be included so the reader can calibrate?

Line 123. Changes evident at dates 13-05-2011 and 08-02-2012: perhaps these are better described as "transient" [changes]?

Line 127 (paragraph). To what extent are these changes simply secondary to abundance?

Line 158 "assemblies". This is the first place in the manuscript that reference is made to the fact that sequence data are assembled: please provide a sentence to cover this?

Line 160. Supp. Figure 2: what do the colours of the circles atop the bars denote? In general figure legends for Supp Figuresneed more detail.

Line 167+. AMR dynamics within Ca. M. parvicella. Line 175 "..confirmed

175 through metatranscriptomic analysis (Figure 2c): I cannot see how Figure 2C supports this statement. Line 179 "…29-22-2011…": I presume you mean 29-11-2011 [the second of the two adjacent points in Figure 2b that show near-zero signal levels?].

Line 185+ Figure 2C. In places, this figure is incomprehensible: can you offset the labels for improved clarity? "…clear and distinct co-occurrence patterns": there is no statistical support for this conclusion provided.

Line 194+. Where are the data that support the genomic "backbone" of these pathogens are detectable in this dataset? Also relevant for Figure 5C. The lack of protein level expression signals (line 316) from these organisms may also be a relevant observation in relation to this point.

Line 222. "…we found that plasmids contributed to an average of 10.8% of all ARGs, while phage contributed to an average of 6.8% of all resistance genes, confirming the general hypothesis that conjugation has the greatest influence on the dissemination of ARGs". Where is the statistical support for this conclusion? Do you think the effect size is scientifically significant?

Line 235. Figure 4B. What multiple correction procedure used and what were the total number of tests corrected for? (see also Figure 5A).

[Editors’ note: further revisions were suggested prior to acceptance, as described below.]

Thank you for resubmitting your work entitled "Mobilome-driven segregation of the resistome in biological wastewater treatment" for further consideration by *eLife*. Your revised article has been evaluated by Bavesh Kana (Senior Editor) and a Reviewing Editor.

The manuscript has been improved but there are some remaining issues that need to be addressed, as outlined below:

Please revise the text to address remaining issues regarding the methodology and statistics, figures, legends and biological aspects to make the document more precise and clear. You will find further details in the reviewers comments below.

*Reviewer #1 (Recommendations for the authors):*

Thank your revision of the paper and responses to my comments, with have addressed many of my concerns. Many parts of the paper are now much clearer. For comments for which I still have concerns, I have listed below, as numbered:

Comment 2.7

In your response, you state "Numbers of taxa are highlighted at the top of each bar, with colors corresponding to the AMR categories on the x-axis.". This will only make sense if these colours are used consistently across figures (which they appear to be, but can you confirm or otherwise?). If so, some reference to that fact should be made.

Comment 2.9

Figure 2C remains challenging. The resolution appears to have increased but the nodes with long annotations are extremely difficult to read, as are the overlapping labels. I don't think it is unreasonable to say that this figure in its present form is not suitable for publication. Have you considered a heatmap (of the adjacency matrix) as an alternative?

Comments 2.12

Firstly I do not see how these tests can be described a two-way ANOVA, given there appears to be only one factor that is tested e.g. phage vs plasmid. Can you confirm that multiple testing has been performed using the total number of tests: that is, the number of ARG categories, encoded (Figure 4) or expressed (Figure 5)? In terms of visualisation, some of the horizontal "significance bars" are solid at both ends, while some show some kind of directionality. Can you clarify what these each mean?

*Reviewer #2 (Recommendations for the authors):*

In this revised version the authors have done a good job of addressing the reviewers' various concerns, such as rephrasing some claims and including additional needed methodological information and statistical support. As consequence, the manuscript and their interpretation of the results are clearer. There are still some small points to be addressed for further clarification of the text.

1. Despite making it clear in the rebuttal that sub-inhibitory antibiotic concentrations can in fact induce HGT, the same does not apply (as far as I know and despite the references shown) for ARGs. Resistant genes can be selected as a consequence of a particular pressure, but I see no indication that these genes actually "facilitate HGT of ARGs into new hosts" as is suggested in the text (L79). Please revise.

2. Is there a possible explanation for the fact that, for example, there were no markers associated with plasmids (or very few) in *Salmonella*, even though plasmids are in fact associated with ARGs? It might also be relevant to mention possible limitations of this approach (as in any approach) since this might help researchers in the future.

3. Lines 496-8 mention the lack of tools to link MGEs and hosts, but the authors should look at the following reference: https://doi.org/10.1101/198713.

4. L422 – In the response to a previous comment, the authors state that the approach can detect "mechanisms novel to a specific microbial reservoir". Since this phrase was not included in the text itself, the sentence is still misleading. They should make clear that when they refer to novel mechanisms, they refer to a specific environment being studied (particularly in L422; perhaps also on L 445).

---

## [Author Response]

[Editors’ note: the authors resubmitted a revised version of the paper for consideration. What follows is the authors’ response to the first round of review.]

Reviewer #1 (Recommendations for the authors):This work aims to provide a broad view of antimicrobial resistance determinants in a wastewater treatment plant by conducting a multi-omics approach on samples taken over 1.5 years. The use of several omics approaches and the distinction of markers associated with either phages or plasmids in the system represent strengths of the study. However, the lack of clarity and methodological details weaken the interpretation and impact of the results.

We recognize the reviewer’s appreciation of the strength of the study as well as their concerns regarding the lack of clarity and methodological details. To address these concerns, we have included additional information on the methodology. We have also thoroughly revised the manuscript to improve clarity as described in detail below.

1. While the multiple omics approach provides valuable insight, it is not entirely clear how expression data adds to what is already known regarding ARGs in the environment and how this expression information sheds new light on the resistome or can be "crucial for monitoring human health conditions" (L412). In this respect it would have been good to see if their findings on ARGs segregating between phages and plasmids correlate with what is known about resistances, at least in the relevant taxa (eg, L 288-9).

We thank the reviewer for this comment. Although metagenomics is useful to characterize the resistome, it does not provide information regarding the actual expression of antibiotic resistance genes (ARGs). In contrast, metatranscriptomics and metaproteomics identifies the ARGs encoded in the metagenome that are actively expressed within the community. For instance, expression indicates that the relevant ARGs are actively expressed as the organisms are exposed to the corresponding antibiotics. As such, linked with metagenomic data, metatranscriptomic data provides a taxonomic, genomic and functional overview of the BWWTP resistome. Thereby potentially reflecting the presence of ARGs in the general human population, as well as the further dissemination in the receiving environment. Moreover, metatranscriptomic data, in addition to metagenomic data, provides essential additional perspectives not only of the short-term adaptation to existing environmental conditions but also regarding selective pressures shaping the resistome.

Regarding the segregation of ARGs according to MGE, while to our knowledge no other study has assessed the differential contribution of MGEs to the dissemination of AMR, various studies have investigated the dissemination of AMR by MGEs in clinically relevant taxa. For instance, correlating with our findings a study by Gupta *et al.* (PMID: 23734150) found that phages are associated with aminoglycoside resistance, among others, in *Salmonella enterica*. Additionally, analogous to our findings, they found that plasmid encoded resistance is heavily associated with pathogenic bacteria such as *S. pneumoniae, S. aureus, K. pneumoniae, E. kobie* and *E. hormaeche.* This has been further highlighted in lines 381-384 in the revised manuscript.

2. Since the results are based on assembled data (L 158), the authors should provide additional and sufficient information regarding these assemblies and how they can be confident of taxa identification to the species level (such as the ESKAPEE group) and the linkage of ARGs to particular microorganisms. This is important, given that much of the analysis and ensuing conclusions are based on linkage of ARGs and MGEs to particular taxa. It would also be important to know how they managed to precisely map expression data to these MGEs and not to chromosomal loci and if perhaps this could also explain some of the discrepancies observed (eg, L397-8).

We thank the reviewer for highlighting this point. Based on this comment, we have restructured and included additional information regarding the methodology in the revised manuscript (lines 457-514). Additionally, we have included an additional figure representing a schematic overview of the methods used (Figure 8).

With respect to the reviewer’s first point relating to this comment, both metagenomic and metatranscriptomic reads were preprocessed and assembled using the Integrated Meta-omic Pipeline (IMP v1.3; available at https://r3lab.uni.lu/web/imp/, PMID: 27986083). Specifically, during processing Illumina Truseq2 adapters were trimmed, and reads of human origin were filtered out, which was followed by a de novo assembly using MEGAHIT (v1.2). For the assembly, both metagenomic and metatranscriptomic reads were co-assembled to increase the contiguity of the assemblies (lines 465-472, in the revised manuscript), in an iterative process to optimize read usage (PMID: 27986083).

The contig-based assembly file was then used as input for the taxonomic classification system Kraken2 which subsequently assigned the bacterial taxonomy, including where relevant the ESKAPEE group, to each contig. To ensure high levels of confidence in taxonomic classification, stringent cutoffs such as a minimum 70% identity of the contig with the database, across 90% of the contig length was used. Since the same contig-based assembly file was used as input for PathoFact, and as such for the prediction of antimicrobial resistance genes and mobile genetic elements, functional- and taxonomic- annotations were linked based on the contig they originated from (lines 487-496, in the revised manuscript).

Finally, to calculate the gene copy number and transcriptome expression, both the metagenomic and metatranscriptomic reads were independently mapped to the assembly files. The tool FeatureCount was then used to extract the raw read counts per contig as well as per individual ORF, as provided by PathoFact (lines 511-514). By using the above-described stringent methods and based on an identical approach in our previous (peer-reviewed) publications (PMIDs: 33597026, 35484157, DOI: 10.1038), we were able to precisely map the expression data to MGEs, including those distinct from the chromosomal loci, with high levels of confidence.

3. It seems that about 18% of all ARGs are associated with MGEs, yet no additional mention is made of the remaining genes, which presumably are chromosomal. Please clarify this in the document and also indicate how this could affect the transcriptomic and proteomic analysis.

We thank the reviewer for highlighting that the remaining resistance genes are found encoded on the bacterial chromosome and have expanded on this in the revised manuscript (lines 226-228).

Regarding the transcriptomic and proteomic analysis, these are not affected at all given that all expressed genes and proteins are taken into account (i.e. both chromosomal and MGE-derived genes).

4. It is intriguing that they observe high levels of peptide resistance in their metatranscriptome (L146) associated with resistance in *E. coli*, which is not a major contributor in the community and the gene was apparently not abundant in their metagenomic analysis. This discrepancy is not fully addressed and is relevant because it can also reflect biases in the approach or conditions of the sites being analyzed.

We thank the reviewer for picking up on this point. While resistance against the peptide antibiotic microcin J25 mediated by YojI has first been described in *E. coli*, we found this resistance gene to be widely distributed among Comamonadaceae members, among others, which are major members in the community. We have now adjusted the manuscript to reflect this fact in lines 149153.

5. It seems that the majority of markers are associated with abundant taxa, which is to be expected, but it is not clear if this is the reason why the authors hypothesize that the majority of ARGs would be in the major taxon M. parvicella (L169 and L175) or why these ARGs would confer fitness advantage (L184), unless of course this correlates with antibiotics present, which was not determined.

We thank the reviewer for highlighting this point. *M. parvicella* generally, as confirmed in previous studies, is found to be the major taxon of the studied wastewater microbial community (PMID: 25424998, 33077707, 28721231). While the mechanisms underlying the abundance of this taxon are numerous, we hypothesized that their success within the WWTP microbial community may also be driven by an increased abundance of antimicrobial resistance genes. Owing to the nature of the sample, i.e. wastewater derived from commercial and anthropogenic influences, we posited that the relatively higher abundance of *M. parvicella* may be driven by the presence of several ARGs, which are likely to be introduced into the WWTP via the intake system. This is further underlined by the expression levels of ARGs encoded by *M. parvicella* indicating a fitness advantage and, thus, allowing it to thrive in the general antibiotic rich environment of a wastewater treatment plant.

6. The document can also benefit from revision of the text, particularly the discussion, to clarify some claims and make the document easier to follow. There are some statements that do not accurately reflect either the scope of the work or the results obtained. Some of these include:L79 – why would ARGs and sub-inhibitory antibiotic concentrations (which should not be lethal) promote HGT?

Various studies have reported both the development of antimicrobial resistance mechanisms as well as the selection of resistant strains (as cited in the manuscript: PMID: 27029295, 20159551, 14506034 and 7726525). Furthermore, there has been further evidence suggesting that subinhibitory concentrations may significantly increase the frequency of dissemination of many types of MGEs (PMID: 21845185, 3710955, 3343220 and 18359835).

L 96 – They do not really shed light on "the evolution" and at best suggest dissemination differences.

The passage has been rephrased as suggested by the reviewer (line 96).

L102 – The analysis suggests but does not necessarily "demonstrate" that MGEs are important drivers of AMR dissemination. In addition, it does not necessarily follow that "assessing the activity of the ARGs" can lead to understanding the underlying mechanisms.

The passage has been corrected as suggested by the reviewer (lines 101-103).

L186 – why population sizes? Did you actually look at association with the size?

We thank the reviewer for highlighting this point and have corrected the passage in the revised manuscript (lines 189-191).

L192 – Is this figure 3a?

This is indeed figure 3a. We thank the reviewer for pointing this out and have adjusted this in the revised manuscript.

On L202 it says that they "assessed the acquisition and dissemination of AMR". The data shows only associations, which is not necessarily acquisition and at best hints at modes of dissemination.

As suggested by the reviewer we have revised the paragraph to highlight the modes of dissemination instead of acquisition.

L292-94- the findings indicate but are not themselves a threat or have potential for dissemination. Please correct.

The paragraph has been corrected as suggested by the reviewer.

L296 – it does not seem that the proteomics validated transcriptomics data, but rather are more of a complementary analysis (L298). Likewise, on L416- It does not seem that proteomics data validated the abundance and expression findings.

The metaproteomic data confirmed the identified resistance mechanisms in the metagenomic data as functional active. However, we agree with the reviewer that the metaproteomic analysis should be seen more as a complementary analysis than validation and have revised the section as suggested.

L413 and 438 – Monitoring based on sequence analysis is based on comparisons using known data and therefore it would be difficult to detect undescribed or novel mechanisms.

As correctly noted by the reviewer, sequence analysis is based on known data and the corresponding databases. Although using machine learning based searches allows us to identify homologs of known antimicrobial resistance genes and mechanisms with current methods, it is still difficult to detect completely novel mechanisms. However, these methods allow for the early detection of mechanisms novel to a specific microbial reservoir. For example, the identified resistance gene *Yoji*, was already reported as a resistance mechanism against microcin J25 in 2005. Nevertheless, resistance was expected to be rare and microcin J25 has even recently been investigated as a potential new commercial antibiotic. Yet, within our BWWTP we not only identified the presence of *Yoji*, heretofore unknown in BWTTPs, we also identified high expression levels of this resistance mechanism.

Reviewer #2 (Recommendations for the authors):In this paper, de Nies and colleagues reanalyse their previously released, ground-breaking multi-omics time series dataset collected from a wastewater treatment plant microbial community over a 14 month period (with 51 sampling days), with the aim of exploring the type and diversity of AMR-related genes present in this microbial community, the type of mobile genetic element they are associated with (plasmid or phage), their association with microbial groups and the nature of their temporal dynamics.This work holds strong interest and utility for workers in the areas of antimicrobial resistance and One Health, and, more broadly, for our understanding the nature of the interconnections between human and natural ecosystems.The starting point is the metagenome assemblies constructed from the collected nucleic acid sequence datasets, and from which contigs form the units of analysis. The analysis is organised around classifying AMR related genes within contig sequences, with particular reference to distinguishing between the type of mobile genetic element (MGE) that harbour them, namely either plasmids or bacteriophage/prophages (assumed to be incorporated into a host genome via transduction). While there are definitive strengths of using the contig-level analysis, I feel far more detail could be provided in the methods description to convince the reader of the validity of the analysis, particularly in relation to drawing associations between MGE sequences and host microbial groups. Exactly how this is done, and the relative strengths and weaknesses, are not fully described and this limits the extent to which the reader can evaluate the overall approach and findings.At certain places in the manuscript, I feel there is an over-reliance of visual data analysis, and more rigorous statistical analysis should be used to test the hypotheses in question; this includes (1) the analysis summarised on line 222 concerning the proportion of ARG/s attributable to plasmids and phages, and (2) the analyses presented in Figure 2C. I note there is also an analysis examining ESKAPEE pathogens, however, there is no direct evidence that the genome "backbone" of these organisms are actually present.Materials and methodsFor an analysis of considerable complexity, the methods section needs more detail and clarity, as it is sometimes hard to understand how a specific result as been obtained.Firstly, the metagenome and metatranscriptome reads from all n=51 samples are co-assembled (line 462-463 ), but it is not clearly stated whether the both sets of data are assembled together?, This appears to be the case, as the reader is given the impression there is a single set of contigs in play? (line 466).

We thank the reviewer for this comment. For the assembly, both metagenomic and metatranscriptomic reads were indeed co-assembled, i.e. assembled together, to increase the contiguity of the assemblies and enhance read usage (PMID: 27986083). Specifically, both metagenomic and metatranscriptomic reads were processed and assembled together in a single set of contigs using the Integrated Meta-omic Pipeline (IMP v1.3; available at https://r3lab.uni.lu/web/imp/). This has been extensively described in the article describing the IMP workflow (PMID: 27986083). However, to clarify this further as pointed out by the reviewer, we have expanded lines 468-473, in the revised manuscript.

Secondly, a critical part of the methodology relates to the annotation of MGE sequences, and the inference of association between MGE sequences and bacterial groups; however. My understanding is that PathoFact will predict whether a contig arises, if classifiable, from either a plasmid or a phage, or whether it derives from a chromosomal genome. The taxonomic annotations for all contig are then made with Kraken. One possibility is that a contig arises from a chromosomal genome and contains an AMR sequence, which seems straightforward, as cognate AMR sequence will be unambiguously associated with the Kraken taxonomic annotation. Whereas another possibility is that the contig arises from a non-chromosomal MGE context; but in that case, it is a little unclear how the taxonomic assignments are actually made (i.e. over lines 477-489); particularly the logic that supports the statement "…to track transmission of AMR between taxa". It is very unclear how these decisions are actually made, but it comes across as rather subjective e.g. "…considering all different predictions of MGEs…". Needless to say, these inferences are critical for the rest of the paper, so can the authors please describe the approach in more detail, including backfilling the underlying scientific logic? A schematic diagram would be useful showing how different sources of data are linked.

We thank the reviewer for this comment. As detailed in our response to comment 1.X. by reviewer 1, we have now provided much more methodological details in the revised manuscript. As indicated by the reviewer, PathoFact does indeed predict whether a contig arises, if classifiable, from either a plasmid or a phage through application of three different prediction methods (PlasFlow, DeepVirFinder and VirSorter). Subsequently, the bacterial taxonomy, such as the ESKAPEE pathogens, are assigned, as highlighted by the reviewer using the taxonomic classification system Kraken2. To ensure confidence in taxonomic classification, stringent cutoffs such as a minimum 70% identity of the contig with the database, across 90% of the contig length was used (lines 487-496). As per the reviewer’s suggestion, and as described in Comment 1.3, to clarify the linkage of different sources of data, a schematic diagram (Supp. Figure 9) has been added to the revised manuscript.

We would further like to clarify that though several tools exist for the identification of plasmids and/or phages, for example: metaplasmidspades, PlasFlow, DeepVirFinder and VIBRANT; tools for comprehensively identifying the respective bacterial hosts for MGEs do not exist. Simultaneously, the hosts for phages can be determined based on classification and linking of CRISPR and Spacer sequences between bacterial and phage sequences, respectively. Since most tools, i.e. SpacePHARER and PHIST, limit this analysis to high-quality viral sequences thus precluding a large fraction of putative identified phages; we did not pursue this approach. The hosts of plasmids, on the other hand, due to the competence of plasmids (e.g. self-replication, broad host range) the precise nature of plasmid ranges are difficult to identify. Therefore, to streamline this process, we used taxonomic associations as a means of linking bacteria with MGEs, given the stringent filtering/identification criteria used. Since we already used stringent cutoffs for plasmid/phage predictions, the original MGE predictions from PathoFact were used asis. This was followed by the taxonomic classification using Kraken2. We have now further clarified these points in the revised manuscripts in lines 493-514.

ResultsFigure 1. What is the level of read coverage that underpins the relative abundance measures used? Can some short statistical summary to cover that be included so the reader can calibrate?

As suggested by the reviewer we have included additional information on the read coverage and depth in the revised manuscript (lines 472-473). We have further included more detailed information including tables or read counts and coverage in our git repository: https://git-r3lab.uni.lu/laura.denies/lao_scripts.

Line 123. Changes evident at dates 13-05-2011 and 08-02-2012: perhaps these are better described as "transient" [changes]?

We thank the reviewer for highlighting this and have revised the description as suggested.

Line 127 (paragraph). To what extent are these changes simply secondary to abundance?

As suggested by the reviewer, the changes in relation to the presence and absence of specific AMR categories may indeed to an extent be secondary to abundance. For instance, as would be expected, highly abundant AMR categories such as genes conferring resistance to aminoglycoside, β-lactam as well as multidrug resistance were found across all timepoints. Meanwhile, lowly abundant AMR categories such as bicyclomycin resistance were found only sporadically within the BWWTP, indicating an association between AMR abundance and prevalence throughout time. However, one should also note the identified exceptions such as fosfomycin resistance, which, although less abundant, was found to be present across all timepoints, demonstrating that even lowly abundant AMR categories may still be prevalent over time.

Line 158 "assemblies". This is the first place in the manuscript that reference is made to the fact that sequence data are assembled: please provide a sentence to cover this?

The paragraph has been revised as suggested (lines 117-121).

Line 160. Supp. Figure 2: what do the colours of the circles atop the bars denote? In general figure legends for Supp Figuresneed more detail.

The different color circles on top of the bars in Supp. Figure 2. denote the different categories of AMR. As suggested by the reviewer, we have adjusted the figure legends in the revised manuscript accordingly.

Line 167+. AMR dynamics within Ca. M. parvicella. Line 175 "...confirmed 175 through metatranscriptomic analysis (Figure 2c): I cannot see how Figure 2C supports this statement. Line 179 "…29-22-2011…": I presume you mean 29-11-2011 [the second of the two adjacent points in Figure 2b that show near-zero signal levels?].

We thank the reviewer for highlighting these points. In agreement with the metagenomic data, ARGs (e.g. aminoglycoside, β-lactam and multi-drug resistance) were found to be highly expressed by *M. parvicella,* likely driven by selective pressure within the antibiotic-rich environment of the BWWTP. We have included an additional supplementary figure to highlight expressions levels of ARGs encoded by *M. parvicella* (Supp. Figure 3 in the revised manuscript). Additionally, we have corrected line 179 (in the revised manuscript) as suggested by the reviewer.

Line 185+ Figure 2C. In places, this figure is incomprehensible: can you offset the labels for improved clarity? "…clear and distinct co-occurrence patterns": there is no statistical support for this conclusion provided.

We thank the reviewer for highlighting this and have adjusted the figure labels. Additionally, we used edges with correlation values of greater than 0.6 correcting for multiple testing where the adjusted p-value <0.05. We have now revised the methods to include this information (lines 565576).

Line 194+. Where are the data that support the genomic "backbone" of these pathogens are detectable in this dataset? Also relevant for Figure 5C. The lack of protein level expression signals (line 316) from these organisms may also be a relevant observation in relation to this point.

We thank the reviewer for the comment. As highlighted in our response to comments 1.2 and 2.2 above, the genomic backbone of the pathogens was based on taxonomic classifications of the contigs using stringent cutoffs. We have now included said data, i.e. taxonomic classification of pathogens and the resistance genes they encode, within our publicly available git repository: https://git-r3lab.uni.lu/laura.denies/lao_scripts.

With respect to the protein level expression, we agree that an overall lack of protein expression is relevant, and therefore now discuss this in detail in the Discussion section in lines 430-433. In this context, there are many factors which affect the detection of proteins, not least the presence of proteases in the WWTP environment. In some cases, the proteins secreted may act as lytic molecules or potentially even degraded. Dreyer *et al.* (https://doi.org/10.1038/s41598-019-478439) reported the proteolytic degradation of bacteriocins, positing the possibility that some of the proteins may be undetectable due to technical reasons. The revised discussion now includes details reflecting these circumstances.

Line 222. "…we found that plasmids contributed to an average of 10.8% of all ARGs, while phage contributed to an average of 6.8% of all resistance genes, confirming the general hypothesis that conjugation has the greatest influence on the dissemination of ARGs". Where is the statistical support for this conclusion? Do you think the effect size is scientifically significant?

We thank the reviewer for highlighting this point. We found the contribution of plasmids contributing to AMR significantly increased compared to phages (p-value <0.05, Two-way ANOVA) and have updated the revised manuscript with the required statistical support (lines 228233).

Line 235. Figure 4B. What multiple correction procedure used and what were the total number of tests corrected for? (see also Figure 5A).

To determine the statistical difference of the results visualized in Figure 4b and Figure 5a, a two-way ANOVA has now been performed, after which the multiple comparison procedure Tukey HSD was performed using the fitted ANOVA as an argument.

[Editors’ note: what follows is the authors’ response to the second round of review.]

The manuscript has been improved but there are some remaining issues that need to be addressed, as outlined below:Please revise the text to address remaining issues regarding the methodology and statistics, figures, legends and biological aspects to make the document more precise and clear. You will find further details in the reviewers comments below.Reviewer #1 (Recommendations for the authors):Thank your revision of the paper and responses to my comments, with have addressed many of my concerns. Many parts of the paper are now much clearer. For comments for which I still have concerns, I have listed below, as numbered:

We thank the reviewer for the recognition of our work and have addressed the remaining concerns in the comments below.

Comment 2.7In your response, you state "Numbers of taxa are highlighted at the top of each bar, with colors corresponding to the AMR categories on the x-axis.". This will only make sense if these colours are used consistently across figures (which they appear to be, but can you confirm or otherwise?). If so, some reference to that fact should be made.

As the reviewer noted the colors corresponding to the AMR categories are indeed used consistently across all figures. As suggested, we have included a reference to this in the figure legends.

Comment 2.9Figure 2C remains challenging. The resolution appears to have increased but the nodes with long annotations are extremely difficult to read, as are the overlapping labels. I don't think it is unreasonable to say that this figure in its present form is not suitable for publication. Have you considered a heatmap (of the adjacency matrix) as an alternative?

We appreciate the reviewer’s concerns regarding Figure 2c. We have considered a heatmap of the adjacency matrix as an alternative, but this resulted in the loss of information provided by the clusters as observed in the network. However, we have revised the original network and adjusted the long annotations as well as the overlapping labels to improve readability. ARGs have been color coded by the corresponding AMR category, with further information regarding specific ARGs provided in a supplementary adjacency matrix (https://git-r3lab.uni.lu/laura.denies/lao_scripts//tree/master/Data). Additionally, the fonts and size of the taxa annotations have been adjusted to improve readability. These adjustments should address the reviewer’s concerns regarding this figure.

Comments 2.12Firstly I do not see how these tests can be described a two-way ANOVA, given there appears to be only one factor that is tested e.g. phage vs plasmid. Can you confirm that multiple testing has been performed using the total number of tests: that is, the number of ARG categories, encoded (Figure 4) or expressed (Figure 5)? In terms of visualisation, some of the horizontal "significance bars" are solid at both ends, while some show some kind of directionality. Can you clarify what these each mean?

We thank the reviewer for highlighting these points. Regarding the two-way ANOVA, the different ARG categories were taken into account as an additional factor. More specifically, in the metagenomic data, a total of 28 ARG categories were tested, of which 6 (aminoglycoside, bacitracin, fosfomycin, MLS, peptide and sulfonamide resistance) were significantly differentially encoded on phages in comparison to plasmids. In the metatranscriptomic data, the same 28 ARG categories were tested of which 6 (aminoglycoside, bacitracin, glycopeptide, mupirocin, peptide and sulfonamide resistance) were significantly differentially expressed on phages versus plasmids. We have clarified this further in the text by further elaborating on the description of the statistical analyses (lines 242-244 and lines 584-586).

Regarding the visualizations, the bars indicating significance should be straight and solid at both ends. We thank the reviewer for pointing out this artifact. We have fixed it in the revised manuscript.

Reviewer #2 (Recommendations for the authors):In this revised version the authors have done a good job of addressing the reviewers' various concerns, such as rephrasing some claims and including additional needed methodological information and statistical support. As consequence, the manuscript and their interpretation of the results are clearer. There are still some small points to be addressed for further clarification of the text.

We thank the reviewer for their appreciation of the revised manuscript. Please see below our responses to the remaining comments.

1. Despite making it clear in the rebuttal that sub-inhibitory antibiotic concentrations can in fact induce HGT, the same does not apply (as far as I know and despite the references shown) for ARGs. Resistant genes can be selected as a consequence of a particular pressure, but I see no indication that these genes actually "facilitate HGT of ARGs into new hosts" as is suggested in the text (L79). Please revise.

We thank the reviewer for highlighting this point and appreciate their concern. Sub-inhibitory concentrations of antibiotics can induce HGT as well as select for genes conferring resistance and/or induce resistant mutations (PMID: 26925045). Additionally, as shown by Datta and Hughes (PMID: 6316165), selection pressures may alter HGT processes, increasing the number of resistance elements which reside on mobile DNA, thereby indirectly facilitating the transmission of said resistance elements to new hosts. However, we appreciate the reviewer’s concerns and have revised the manuscript to clarify and elaborate on these aspects (lines 78-83).

2. Is there a possible explanation for the fact that, for example, there were no markers associated with plasmids (or very few) in Salmonella, even though plasmids are in fact associated with ARGs? It might also be relevant to mention possible limitations of this approach (as in any approach) since this might help researchers in the future.

We thank the reviewer for bringing up this point. While *Salmonella* was identified throughout the time series in the BWWTP, the overall levels of detection for this genus were quite low (relative abundance of 0.01-0.02%). In addition, resistance genes encoded on the *Salmonella* genome were only observed at 5 timepoints, which may be related to the low abundances within this BWWTP. This may also be the reason that, any resistance-encoding plasmids associated with *Salmonella* were undetected.

As highlighted by the reviewer in comment #3 (below), there are Hi-C based methods which provide the capacity to link MGEs and hosts during sample preparations and subsequent sequencing (PMID: 19815776 and https://doi.org/10.1101/198713). We have now included this aspect in the Discussion in lines 410-418. Other methods include targeted sequencing of plasmids of interest using the methodology established by Li et al. (PMID: 29325009), which has been highlighted in the revised Discussion section in lines 402-409.

3. Lines 496-8 mention the lack of tools to link MGEs and hosts, but the authors should look at the following reference: https://doi.org/10.1101/198713.

We thank the reviewer for highlighting this point. While Hi-C-based methods allow capturing of inter-chromosomal junctions, such as plasmid-genome interactions, it requires a different processing of the samples prior to sequencing to induce covalent linkage among DNA molecules. As such we did not include said methods in our manuscript. However, we agree with the reviewer that Hi-C-based methods provide an important method to link MGEs and hosts and have rephrased the corresponding section accordingly (lines 410-418).

4. L422 – In the response to a previous comment, the authors state that the approach can detect "mechanisms novel to a specific microbial reservoir". Since this phrase was not included in the text itself, the sentence is still misleading. They should make clear that when they refer to novel mechanisms, they refer to a specific environment being studied (particularly in L422; perhaps also on L 445).

We thank the reviewer for highlighting this point and have now revised the paragraph as suggested (lines 439-441 and lines 465-468).